# FAIR CONFORMAL CLASSIFICATION VIA LEARNING REPRESENTATION-BASED GROUPS

**Senrong Xu**[1], **Yanke Zhou**[1], **Yuhao Tan**[1], **Zenan Li**[2], **Yuan Yao**[1], **Taolue Chen**[3],
**Feng Xu**[1], **Xiaoxing Ma**[1]

[1] State Key Lab of Novel Software Technology, Nanjing University, China,
[2] ETH Zürich, [3] Birkbeck, University of London
`{srxu,yankezhou,yhtan}@smail.nju.edu.cn, zenan.li@inf.ethz.ch`
`{y.yao,xf,xxm}@nju.edu.cn, t.chen@bbk.ac.uk`

## ABSTRACT

Conformal prediction methods provide statistically rigorous marginal coverage guarantees for machine learning models, but such guarantees fail to account for algorithmic biases, thereby undermining fairness and trust. This paper introduces a fair conformal inference framework for classification tasks. The proposed method constructs prediction sets that guarantee conditional coverage on adaptively identified subgroups, which can be implicitly defined through nonlinear feature combinations. By balancing effectiveness and efficiency in producing compact, informative prediction sets and ensuring adaptive equalized coverage across unfairly treated subgroups, our approach paves a practical pathway toward trustworthy machine learning. Extensive experiments on both synthetic and real-world datasets demonstrate the effectiveness of the framework.

## 1 INTRODUCTION

The rapid advancement of modern machine learning models, especially deep neural networks, has enabled their deployment in high-stake decision-making situations such as medical diagnoses (Kaur et al., 2020), resume filtering (Deshpande et al., 2020), and financial fraud detection (Kamuangu, 2024). Despite their strong average performance, real-world deployment raises critical challenges, notably in uncertainty quantification (Guo et al., 2017; Ahmed et al., 2023) and algorithmic fairness (Berk et al., 2024; Almasoud & Idowu, 2025).

Ensuring reliable decision-making necessitates the development of unbiased uncertainty measures, as even highly accurate models are prone to producing over-confident and erroneous predictions (Ovadia et al., 2019). Conformal Prediction (CP, (Vovk et al., 2005; Smith, 2024)) has emerged as a key framework for providing distribution-free, model-agnostic prediction sets with user-specified (marginal) coverage guarantees. These sets provide reliable uncertainty information for decision-makers especially when the set size is small (i.e., with high efficiency).

On the other hand, algorithmic biases often manifest as disproportionately poor performance on the subgroup defined by specific feature conditions (e.g., *Race=Black & Gender=Female*), which may arise from imbalanced data distribution or model inherent limitations (Hellman, 2020). These biases underscore the need for algorithmic fairness mechanisms that extend beyond average performance to ensure equitable treatment across all groups (Fabris et al., 2022; Das et al., 2023). However, there may exist tensions between the efficiency of CP and algorithmic fairness, because the former desires a small prediction set, while the latter may necessitate larger sets for equal conditional coverage across all subgroups (Gibbs et al., 2025).

Conformal prediction with *equalized coverage* (Romano et al., 2020a) provides a pragmatic approach to the efficiency–fairness trade-off. This approach ensures that the target coverage level (e.g., 90%) is satisfied not only marginally over the entire population, but also conditionally on each protected group of interest. However, acquiring prediction sets with equalized coverage is challenging, as the number of all plausible groups of interest is exponential in the number of features. A

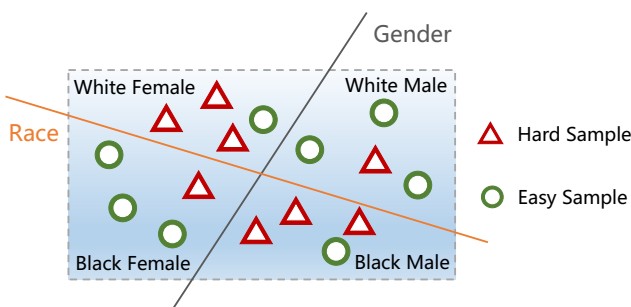

Figure 1: An illustrative example. The group space is divided into four parts by the feature *Race* and *Gender*. Hard samples (red triangles) are unfairly treated by the classifier, and easy samples (green circles) are normally treated. Note that a single feature (either Race or Gender) cannot discover the unfair subgroups (both have four triangles and four circles). Stronger expressiveness is desirable to capture the unfair subgroup "*White Female* or *Black Male*".

straightforward enumeration is practically infeasible both statistically and computationally, especially on multi-dimensional (continuous) features. Indeed, Romano et al. (2020a) only takes each single feature as the condition of groups (e.g., a group defined by *Gender=Female*), which is an arguably insufficient representation of the entire space of groups.

Later, Zhou & Sesia (2024) observe that algorithmic biases often concentrate on a minority of subgroups, and propose adaptively fair conformal prediction (AFCP) to identify these potentially disadvantaged subgroups. In a nutshell, AFCP computes the conditional coverage score for each discrete feature and selects the top-$k$ sensitive features with a greedy strategy (where $k$ is a hyperparameter). However, this group identification method still has limited expressiveness. For example, it cannot capture groups defined by a nonlinear combination of features, such as Exclusive OR (see the subgroup "*White Female* or *Black Male*" in Fig. 1). Additionally, AFCP is based on Naïve Bayes, which incurs a high computational cost and restricts its applicability to continuous features.

## 1.1 OUR CONTRIBUTIONS

In this paper, we propose a new group-fair conformal prediction method, **fair conformal prediction for representation-based groups** (FAREG), which accommodates both group expressiveness and time efficiency. Different from existing work (Romano et al., 2020a; Zhou & Sesia, 2024) which directly extracts groups from the raw input feature $X$, our approach encodes $X$ into a latent representation $Z$ via a mapping $Z = f(X)$, and learns unfair groups characterized by the low group coverage based on $Z$. The introduction of $Z$ as a high-level representation of features strengthens the expressiveness of models, allowing a thorough exploration of groups. Meanwhile, we can enhance the interpretability by reconstructing input $X$ from the encoding $Z$. To this end, we carefully design an encoder-decoder architecture and the optimization objective, based on the principle of variational inference.

In addition, we propose a *nonlinear* version of the conditional coverage metric WSC (Cauchois et al., 2021), namely WSC$^+$, aiming to evaluate the conditional coverage of unfairly treated groups more precisely. This allows users to check a conformal procedure and to compare multiple alternative conformal procedures.

The main contributions of this paper are summarized as follows. First, we propose a new conformal prediction method to enhance the expressiveness of unfair group identification. Second, we extend the traditional conditional coverage metric WSC to a nonlinear version WSC$^+$ for more accurate evaluation. Comprehensive experiments on both synthetic and real-world datasets confirm the effectiveness and efficiency of our proposed method.

## 2 PRELIMINARY

For any natural number $n$, we write $[n] := \{1, \ldots, n\}$. We work with the most widely-used version of conformal prediction, i.e., *split* conformal prediction, where we assume a calibration set

$\mathcal{D} = \{(X_i, Y_i)\}_{i=1}^N$ of i.i.d. (or simply exchangeable) observations sampled from an (unknown) distribution $P_{XY}$. In standard classification, $X_i \in \mathcal{X}$ represents the input feature from a feature space $\mathcal{X} \subseteq \mathbb{R}^d$ and $Y_i \in [L]$ is a categorical label. A given classifier $\hat{f}$ is trained (on a training set) to predict the conditional distribution $P(Y \mid X)$. Furthermore, $X_{N+1}$ is a test instance with an unknown label $Y_{N+1}$ sampled by $P_{XY}$. CP constructs a prediction set $C(X_{N+1})$ for $Y_{N+1}$ based on $\mathcal{D}$. The output $C(X_{N+1})$ guarantees marginal coverage at a user-specified level, i.e.,

$$\mathbb{P}[Y_{N+1} \in C(X_{N+1})] \geq 1 - \alpha,$$

where $\alpha \in (0, 1)$ is a predefined miscoverage rate.

Typically, CP proceeds in three steps: (1) computing the predefined conformity score $V(x_i, y_i)$ for each sample $(x_i, y_i) \in \mathcal{D}$ using the predictive results of the classifier $\hat{f}$; (2) setting $(1-\alpha)(1+1/N)$-quantile score of $\mathcal{D}$ as a threshold $\hat{\eta}$; (3) constructing the prediction set $C_{\mathfrak{m}}(X_{N+1}, \mathcal{D}) := \{y \in [K] \mid V(X_i, y) \geq \hat{\eta}\}$, which is used as $C(X_{N+1})$ for $X_{N+1}$.

It can be shown that $C_{\mathfrak{m}}(X_{N+1}, \mathcal{D})$ meets the desirable *marginal coverage*. Intuitively, marginal coverage implies that the prediction set is guaranteed to contain the true label with the *average* $1-\alpha$ probability over the population. However, this guarantee is deemed to be insufficient, especially when miscoverage exhibits systematic bias, disproportionately affecting individuals belonging to groups characterized by certain features.

By contrast, *conditional coverage* requires $\mathbb{P}[Y_{N+1} \in C(X_{N+1}) \mid X_{N+1} = x] \geq 1 - \alpha$ for each $x \in \mathcal{X}$. This is much stronger as it demands correct coverage across all regions of the feature space, not just on average. However, achieving conditional coverage is impossible without imposing extra assumptions on the underlying distribution $P_{XY}$ (such as the smoothness of $P_{XY}$ (Cai et al., 2014; Lei & Wasserman, 2014) and strictly limiting the size of feature space $\mathcal{X}$ (Lee & Barber, 2021)). As these strong assumptions are often violated, conditional coverage is less meaningful in practice.

Equalized coverage (Romano et al., 2020a) represents a pragmatic compromise to ensure validity across *predefined* sample groups that need to be protected. Given a group $\mathcal{G} \subseteq \mathcal{X}$, it is required that

$$\mathbb{P}[Y_{N+1} \in C(X_{N+1}) \mid X_{N+1} \in \mathcal{G}] \geq 1 - \alpha$$

for all $\mathcal{G}$ of interest. In particular, these groups are typically related to some specific features called sensitive features.

However, the requirement for rigorous equalized coverage is localized, as algorithmic biases disproportionately affect only a minority of subgroups (Zhou & Sesia, 2024), as mentioned in Seciton 1. Therefore, AFCP further proposes adaptive equalized coverage based on equalized coverage, formalized by

$$\mathbb{P}[Y_{N+1} \in C(X_{N+1}) \mid X_{N+1} \in \hat{\mathcal{G}}] \geq 1 - \alpha, \tag{1}$$

where $\hat{\mathcal{G}}$ is adaptively selected corresponding to sensitive features. Eq. 1 indicates that $C(X_{N+1})$ is well-calibrated for the selected group $\hat{\mathcal{G}}$ defined by these sensitive features.

## 3 METHODOLOGY

This section presents FAREG, a learning-based method that adaptively identifies groups affected by algorithmic bias and adjusts their prediction sets to achieve equalized coverage while preserving high informativeness.

### 3.1 LEARNING REPRESENTATION-BASED GROUPS

**Optimization Objective.** For any feature $x \in \mathcal{X}$, we write its encoding $z = f(x) \in \mathcal{Z}$, where $\mathcal{Z}$ is a latent representation space. Intuitively, $z$ denotes the latent representations of feature combinations of $x$. We introduce a random binary variable $S$ and $Z$ taking values in $\mathcal{Z}$ to formalize the membership of a group. Naturally, we consider a conditional distribution $P(S \mid Z)$ such that the probability of $x \in \hat{\mathcal{G}}$ for a group $\hat{\mathcal{G}}$ is equal to $\mathbb{P}(S = 1 \mid Z = f(x))$. Our goal is twofold: (1) to learn an encoding $Z = f(X)$ that is maximally informative about $S$ and $X$, while (2) $Z$ does not reveal the identity of any individual $i$ in the sample (e.g., the calibration set).

We apply the *deep variational information bottleneck* (Deep VIB) method (Alemi et al., 2017). Specifically, for two random variables $X$ and $Y$ with the joint pdf (parameterized by $\theta$), $p_\theta(x, y)$, $I(X, Y; \theta) = \int p_\theta(x, y) \log \frac{p_\theta(x,y)}{p_\theta(x)p_\theta(y)} \, dxdy$ denotes their mutual information. The optimization objective can be formalized as

$$\max I(Z, S; \theta) + I(Z, X; \theta) - \beta I(Z, i; \theta),$$

where $i$ is a random variable to take any instance from the sample (e.g., in this paper, the calibration set $\mathcal{D}$) with a uniform distribution, $\theta$ is the model parameter, and $\beta$ is a weight hyperparameter. (We abbreviate $I(Z, S), I(Z, X), I(Z, i)$ as $I_1, I_2, I_3$ for convenience.)

By introducing $q_\phi(s|z), q_\varphi(x|z), r(z)$ as the variational approximation to $p_\theta(s|z), p_\theta(x|z), p(z)$ in respective terms, we perform variational inference and obtain

$$I_1 + I_2 - \beta I_3 \geq \int p_\theta(x)p_\theta(s\,|\,x)p_\theta(z\,|\,x) \log q_\phi(s\,|\,z) \, \mathrm{d}x \, \mathrm{d}s \, \mathrm{d}z$$
$$+ \int p_\theta(x)p_\theta(z\,|\,x) \log q_\varphi(x\,|\,z) \, \mathrm{d}x \, \mathrm{d}z - \frac{\beta}{N} \sum_i \int p_\theta(z\,|\,x_i) \log \frac{p_\theta(z\,|\,x_i)}{r(z)} \, \mathrm{d}z.$$

(The details are given in Appendix A.1.)

In practice, we can approximate $p_\theta(x, s) = p_\theta(x)p_\theta(s|x)$ and $p_\theta(x)$ using the empirical distribution on the observations (e.g., the calibration set $\mathcal{D}$). As for $p_\theta(z|x)$, the reparameterization trick (Kingma & Welling, 2013) forces $z$ to conform to a normal distribution which relies on $x_i$, and hence its deterministic function can be rewritten as $z = f(x, \epsilon)$ with an (auxiliary) noise variable $\epsilon$.

Substituting all of these into the above equation, we obtain the following loss function

$$\mathcal{L} = -\frac{1}{N} \sum_{i=1}^{N} \left( \mathbb{E}_{\tilde{z} \sim f(x_i, \epsilon)}[\log q_\phi(s_i\,|\,\tilde{z}) + \log q_\varphi(x_i\,|\,\tilde{z})] - \beta D_{\mathrm{KL}}(p_\theta(z\,|\,x_i) \| r(z)) \right). \qquad (2)$$

Intuitively, the expected log-likelihood $\mathbb{E}_{\tilde{z} \sim f(x_i, \epsilon)}[\log q_\phi(s_i\,|\,\tilde{z}) + \log q_\varphi(x_i\,|\,\tilde{z})]$ allows the encoding $\tilde{z}$ to predict $s_i$ and regenerate $x_i$ simultaneously, whereas the Kullback-Leibler (KL) divergence aims to compress the remaining useless information of $\tilde{z}$.

**Instantiation.** Eq. 2 suggests a natural design of the Encoder-Decoder architecture. In our method, the stochastic encoder with parameter $\theta$ has the form $p_\theta(z\,|\,x) = \mathcal{N}(z\,|\,f_\mu(x), f_\sigma(x))$, where $f_\mu(x)$ and $f_\Sigma(x)$ are two MLP networks to output the mean and variance of a normal distribution. We set $r(z)$ as a standard normal distribution $\mathcal{N}(0, 1)$ and directly minimize the KL divergence term in Eq. 2 using the reparameterization trick.

We now concentrate on two decoders with parameters $\phi$ and $\varphi$. The instantiation of decoder with parameter $\varphi$ is trivial. For the expected log-likelihood $\mathbb{E}_{\tilde{z} \sim f(x_i, \epsilon)}[\log q_\varphi(x_i\,|\,\tilde{z})]$ in Eq. 2, we utilize the standard Mean Squared Error (MSE) as the reconstruction loss (Kingma & Welling, 2013).

Decoder with parameter $\phi$ aims at predicting $S$, which indicates whether the sample $X$ belongs to group $\hat{\mathcal{G}}$ or not. Assume a set of observations, e.g., the calibration set $\mathcal{D} = \{(X_i, Y_i)\}_{i=1}^N$. The distribution $P(S\,|\,X)$ can be viewed as a binary classifier $h$ comprising an encoder with parameter $\theta$ and a decoder with the parameter $\phi$. The result of $h$ on $\mathcal{D}$ is a vector $\mathbf{s} = [s_1, \ldots, s_N] \in \{0, 1\}^N$. Let $\hat{\mathcal{G}}_\mathbf{s} \subseteq \mathcal{D}$ denote the group determined by $\mathbf{s}$ on $\mathcal{D}$ and $\mathcal{H}$ be the family of all plausible $h$. We extend an inequality (Cauchois et al., 2021) to measure the deviation between the empirical coverage probability $\mathbb{P}_n$ on $\mathcal{D}$ and the oracle coverage probability $\mathbb{P}$.

**Proposition 1.** *Let the VC-dimension $VC(\mathcal{H}) \leq R$ and $\delta = |\hat{\mathcal{G}}_\mathbf{s}|/N$ be the proportion of $\hat{\mathcal{G}}_\mathbf{s}$ to the entire dataset. Then the gap between the empirical coverage probability $\mathbb{P}_n$ on the observations and the oracle coverage probability $\mathbb{P}$ is upper bounded, i.e., there exists some constant $C_1$ for all $\tau > 0$*

$$\sup_{h \in \mathcal{H}} \left\{ |\mathbb{P}_n[Y \in C(X)\,|\,X \in \hat{\mathcal{G}}_\mathbf{s}] - \mathbb{P}[Y \in C(X)\,|\,X \in \hat{\mathcal{G}}_\mathbf{s}]| \right\} \leq C_1 \sqrt{\frac{R \log N + \tau}{\delta N}}$$

*holds with probability at least $1 - e^{-\tau}$.*

Proposition 1 (cf. Appendix A.2 for proof) highlights two key directions for reducing the discrepancy between $\mathbb{P}_n$ and $\mathbb{P}$. First, a lower VC-dimension $VC(h)$ leads to a more precise estimation $\mathbb{P}_n$,

implying that the classifier $h$ should exhibit limited complexity. Second, the selected group must be sufficiently large to ensure reliable estimation.

We maximize the expected log-likelihood $\mathbb{E}_{\tilde{z} \sim f(x_i, \epsilon)}[\log q_\phi(s_i \mid \tilde{z})]$ in Eq. 2 via minimizing the expected empirical conditional coverage of the selected group $\hat{\mathcal{G}}$. The group $\hat{\mathcal{G}}$ on $\mathcal{D}$ is determined by a random vector $\mathbf{S}$, sampled from a joint Bernoulli distribution $B = \prod_{i=1}^N \text{Bernoulli}(q_\phi(S_i = 1 \mid \tilde{z}))$. Hence, given $\mathcal{D}$, we formulate the following optimization problem :

$$\min_\phi \mathbb{E}_{\mathbf{S} \sim B}[\mathbb{P}_n[Y \in C(X) \mid X \in \hat{\mathcal{G}}_{\mathbf{S}}]] \quad \text{s.t.} \quad \frac{1}{N} \sum_{i=1}^N q_\phi(S_i = 1 \mid \tilde{z}) \geq \delta. \tag{3}$$

In the above minimization problem, $\delta = |\hat{\mathcal{G}}_{\mathbf{s}}|/N$ denotes the the proportion of the selected group size to the whole dataset $\mathcal{D}$, and the decoder with parameter $\phi$ is a simple logistic regression model of the form $q_\phi(s \mid \tilde{z}) = \sigma(s \mid f_m(\tilde{z}))$, where $\sigma$ is the sigmoid function and $f_m$ is a MLP network.

To solve the constrained optimization problem, we employ the Projected Gradient Descent (PGD), an iterative optimization algorithm (Madry et al., 2017), to optimize the parameter $\phi$. In each training step, PGD performs a gradient descent update and then projects the new point onto the feasible set to ensure all constraints are satisfied. Specifically, when the predictive distribution $q_\phi(s \mid \tilde{z})$ does not meet the constraint $\frac{1}{N} \sum_{i=1}^N q_\phi(S_i = 1 \mid \tilde{z}) \geq \delta$ after one back propagation process, we project it back onto the constraint-friendly space. Such a projection is equivalent to an $\ell_2$ distance minimization problem. Let $q_\phi^*(s_1 \mid \tilde{z}) \geq \cdots \geq q_\phi^*(s_N \mid \tilde{z})$ be the descending order of $\{q_\phi(s_i \mid \tilde{z})\}_{i=1}^N$, and the projection results in

$$q_\phi'(s_i \mid \tilde{z}) = \min\left(1, q_\phi(s_i \mid \tilde{z}) + \frac{\omega}{2}\right), \tag{4}$$

where $\omega = (\delta - k - \sum_{i=k+1}^N q_\phi^*(s_i \mid \tilde{z}))/(N-k) \geq 0$, $k \in [N]$ is the greatest index to satisfy $q_\phi^*(s_k \mid \tilde{z}) + \omega/2 \geq 1$ and $q_\phi^*(s_{k+1} \mid \tilde{z}) + \omega/2 < 1$. (The details are given in Appendix A.3.)

Overall, we employ the empirical conditional coverage loss $\mathcal{L}_{\text{CC}}$, the reconstruction loss $\mathcal{L}_{\text{MSE}}$, and the KL divergence loss $\mathcal{L}_{\text{KL}}$ to replace the corresponding terms in Eq. 2, resulting in

$$\mathcal{L} = \mathcal{L}_{\text{CC}} + \mathcal{L}_{\text{MSE}} - \beta \mathcal{L}_{\text{KL}}. \tag{5}$$

## 3.2 Constructing the Adaptive Prediction Sets

After selecting the unfair group $\hat{\mathcal{G}}$, we proceed to construct the final prediction set with $\hat{\mathcal{G}}$. First, a standard conformal prediction set $C_{\mathfrak{m}}(X_{N+1}, \mathcal{D})$ is constructed using classic adaptive conformal prediction. Then, we perform $T$ sampling of the vector $\mathbf{s}_t$ ($t \in [T]$) from the joint Bernoulli distribution $B$ learned by models in Eq. 3. Each $\mathbf{s}_t$ defines a group $\hat{\mathcal{G}}_{\mathbf{s}_t}$, and such group is used as a calibration set to build a prediction set $C_{\mathfrak{m}}(X_{N+1}, \hat{\mathcal{G}}_{\mathbf{s}_t})$ as mentioned in Section 2. The final prediction set for $Y_{N+1}$ is given by the union of all these sets:

$$C(X_{N+1}) = C_{\mathfrak{m}}(X_{N+1}, \mathcal{D}) \cup \bigcup_{t=1}^T C_{\mathfrak{m}}(X_{N+1}, \hat{\mathcal{G}}_{\mathbf{s}_t}). \tag{6}$$

Our approach FAREG is summarized in Algorithm 1. To analyze its time complexity, assume we have $M$ test instances and the complexity of conducting classic conformal prediction is $\mathcal{O}(N+M)$. Then, training the model to select groups is $\mathcal{O}(EN(|\theta| + |\phi| + |\varphi|))$, where $E$ is the number of epochs. For all $M$ test instances, the time of selecting groups and constructing prediction sets is $\mathcal{O}(TN + TM)$. The overall complexity of our FAREG is $\mathcal{O}(EN(|\theta| + |\phi| + |\varphi|) + T(N + M))$, which is $\mathcal{O}(N + M)$, disregarding constant multiplicative factors. In contrast, the complexity of AFCP is $\mathcal{O}(N \log N + NM)$ (Zhou & Sesia, 2024).

The following result, proved in Appendix A.4, ensures that the prediction set $C(X_{N+1})$ generated by FAREG achieves adaptive equalized coverage (Eq. 1) over the selected group set $\{\hat{\mathcal{G}}_{\mathbf{s}_t}\}_{t=1}^T$.

**Theorem 1.** *If $\{(X_i, Y_i)\}_{i=1}^{N+1}$ are exchangeable, the prediction set $C(X_{N+1})$ and the selected group set $\{\hat{\mathcal{G}}_{\mathbf{s}_t}\}_{t=1}^T$ output by Algorithm 1 satisfy the adaptive equalized coverage defined in Eq. 1, and this guarantee still holds when the selected groups are defined by a more complex combination of features (e.g., non-linear) compared to AFCP.*

---

**Algorithm 1** The overall framework of FAREG.

---

**Input:** calibration dataset $\mathcal{D} = \{X_i, Y_i\}_{i=1}^N$; test instance with feature $X_{N+1}$; list of $K$ sensitive features; pre-trained classifier $\hat{f}$; fixed rule to compute nonconformity scores; level $\alpha \in (0, 1)$; selected group size proportion $\delta$; hyperparameter $\beta$; sampling times $T$;

**Output:** prediction set $C(X_{N+1})$; selected group set $\{\hat{\mathcal{G}}_{\mathbf{s}_t}\}_{t=1}^T$.

1: Construct classic conformal prediction set $C_{\mathfrak{m}}(X_{N+1}, \mathcal{D})$ based on the output of $\hat{f}$;
2: **for** each batch **do**
3:     Calculate KL divergence loss $\mathcal{L}_{KL}$ with reparameterization trick;
4:     Sample $\tilde{z} \sim f(x, \epsilon)$;
5:     Calculate conditional coverage loss $\mathcal{L}_{CC}$ and reconstruction loss $\mathcal{L}_{MSE}$ using $\tilde{z}$;
6:     Put all losses together in $\mathcal{L}$ as defined in Eq. 5;
7:     Update parameters $\theta, \phi$ and $\varphi$ via the gradient descent of $\mathcal{L}$;
8:     **if** $\sum_{i=1}^N q_\phi(S_i = 1 \,|\, \tilde{z}) < \delta \cdot N$ **then**
9:         Project each $q_\phi(S_i = 1 \,|\, \tilde{z})$ to satisfy minimum set constraint using Eq. 4;
10:     **end if**
11: **end for**
12: **for** $t \in [T]$ **do**
13:     Sample $\mathbf{s}_t \sim B$;              ▷ *B is a joint Bernoulli distribution mentioned in Eq. 3*
14:     Construct $C_{\mathfrak{m}}(X_{N+1}, \hat{\mathcal{G}}_{\mathbf{s}_t})$;
15: **end for**
16: Construct prediction set $C(X_{N+1})$ following Eq. 6.

---

## 4 EXPERIMENTS

### 4.1 EXPERIMENTAL SETUP

**Baselines.** We select the classic CP method Marginal (Romano et al., 2020b) for classification, the initial CP method Partial (Romano et al., 2020a) considering equalized coverage, and the state-of-the-art method AFCP (Zhou & Sesia, 2024) as our baselines. The vanilla version of AFCP is designed to pick at most one sensitive feature (referred to as AFCP1). We also extend AFCP1 to select two sensitive features (referred to as AFCP2), given unreal, strong prior knowledge. Note that in real-world applications, it is typically unknown exactly how many features the unfair group may correspond to.

**Evaluation Metrics.** To evaluate the prediction sets $C(X_{N+1})$ produced by different CP methods, we use the coverage conditional on a specific group (referred to as Group Coverage), Average Coverage (viz., marginal coverage), and Average Size (viz., efficiency) as the metrics.

Additionally, we propose a new conditional coverage metric, viz., WSC$^+$, to capture groups defined by complicated (nonlinear) feature relationships. Traditional conditional coverage metric (Cauchois et al., 2021) considers the worst coverage over all slabs containing $\delta$ mass on the observations, which is defined as

$$\text{WSC}_n(C, \mathbf{v}) := \inf_{a < b} \left\{ \mathbb{P}_n(Y \in C(X) \,|\, a \le \mathbf{v}^T X \le b) \ \text{s.t.} \ \mathbb{P}_n(a \le \mathbf{v}^T X \le b) \ge \delta \right\},$$

where $\mathbf{v} \in \mathbb{R}^d$ and $a < b \in \mathbb{R}$.

To strengthen the WSC metric, we replace the linear mapping $\mathbf{v}^T$ in the above equation with an arbitrary non-linear function $\pi$, giving rise to WSC$^+$, i.e.,

$$\text{WSC}_n^+(C, \pi) := \inf_{a < b} \left\{ \mathbb{P}_n(Y \in C(X) \,|\, a \le \pi(X) \le b) \ \text{s.t.} \ \mathbb{P}_n(a \le \pi(X) \le b) \ge \delta \right\}. \tag{7}$$

Assume a quadratic function $\pi(\mathbf{x}) = \mathbf{x}^T \mathbf{W} \mathbf{x} + \mathbf{v}^T \mathbf{x}$, where $\mathbf{W} \in \mathbb{R}^{d \times d}$ and $\mathbf{v} \in \mathbb{R}^d$. We uniformly draw 1,000 samples $\pi_j = \{\mathbf{W}_j, \mathbf{v}_j\}$ to compute the worst-slab coverage for each $\pi_j$ on the test instances. Following Cauchois et al. (2021), we use the grid search to achieve the optimal $a, b$ satisfying the desiderata as well. In this case, we have a lower bound for our metric WSC$^+$.

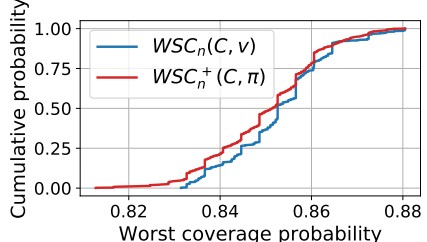

Figure 2: CDF of Conditional Coverage ($\delta = 0.5$), which plots the respective cumulative probability curves of different worst-slab coverage discovered by $\text{WSC}_n(C, \mathbf{v})$ and $\text{WSC}_n^+(C, \pi)$ over 1,000 samplings. The red curve is always above the blue curve, indicating that our $\text{WSC}_n^+(C, \pi)$ finds more groups with the poor coverage than $\text{WSC}_n(C, \mathbf{v})$.

Table 1: Performance of $\text{WSC}_n$ and $\text{WSC}_n^+$ metrics w.r.t. different $\delta$. We repeat the experiment 10 times, and report the average results (the value in () is the standard deviation). Smaller coverage is better. Our metric $\text{WSC}_n^+$ performs better than $\text{WSC}_n$ by up to 9.89% to mine the group with the minimum worst-slab coverage (defined in Eq. 8).

| METRIC | $\delta = 0.1$ | $\delta = 0.2$ | $\delta = 0.3$ | $\delta = 0.4$ | $\delta = 0.5$ |
|---|---|---|---|---|---|
| $\text{WSC}_n$ | 0.616 (0.053) | 0.748 (0.037) | 0.793 (0.025) | 0.822 (0.023) | 0.842 (0.023) |
| $\text{WSC}_n^+$ | 0.582 (0.047) | 0.674 (0.048) | 0.750 (0.034) | 0.800 (0.028) | 0.829 (0.024) |
| IMP. | -5.52% | -9.89% | -5.42% | -2.68% | -1.54% |

**Proposition 2.** *Let $\pi(\mathbf{x}) = \mathbf{x}^T \mathbf{W} \mathbf{x} + \mathbf{v}^T \mathbf{x}$ be a quadratic function and $\Pi$ be a parameter space of $\pi$. Then, if $C$ effectively provides conditional coverage at level $1 - \alpha$, we have*

$$WSC_n^+ = \inf_{\pi \in \Pi} WSC_n^+(C, \pi) \geq 1 - \alpha - \mathcal{O}(1)\sqrt{\frac{\mathcal{O}(d^2)\log N}{\delta N}}. \tag{8}$$

The proof is given in Appendix A.5.

To demonstrate the advantages of the new metric $\text{WSC}^+$, we randomly draw the features $X \in [0, 1]^{10}$ from a uniform distribution and create a simple dataset for classification as described in Appendix B.1. Note that we define the group needed to be protected to satisfy $(X[0] \geq 0.1) \oplus (X[1] \geq 0.1) = \text{True}$. We respectively plot the Cumulative Distribution Functions (CDF) of $\text{WSC}_n(C, v)$ and $\text{WSC}_n^+(C, \pi)$ over 1,000 samples $\pi_j$ when $\delta = 0.5$ in Fig. 2, and observe that our $\text{WSC}_n^+(C, \pi)$ always reveals the groups with the worse coverage than that of $\text{WSC}_n(C, v)$, which can be attributed to representational capability of the nonlinear function $\pi$ in $\text{WSC}_n^+(C, \pi)$.

Moreover, we also list the average results of two metrics, $\text{WSC}_n$ and $\text{WSC}_n^+$, as $\delta$ increases over 10 repeated experiments in Table 1. Similar to Fig. 2, the minimum worst-slab coverage found by our $\text{WSC}_n^+$ is smaller than that found by $\text{WSC}_n$ by up to 9.89%. As $\delta$ increases, the condition coverage tends to the marginal coverage (0.9), and the gap between $\text{WSC}_n$ and $\text{WSC}_n^+$ narrows, as expected.

**Implementations.**[1] All the experiments are carried out on NVIDIA GeForce RTX 3090. We repeat each experiment 10 times and report the average to suppress randomness. We set $\delta = 0.5$ for $\text{WSC}_n^+$ by default. More implementation details, such as hyperparameters and training settings, are presented in Appendix B.1.

## 4.2 SYNTHETIC DATA

We evaluate our method on synthetic data designed to mimic a mental illness diagnosis scenario. The dataset includes six possible labels: Depression, Anxiety Disorders, Bipolar Disorder, Schizophrenia, Anorexia, and Post-Traumatic Stress Disorder (PTSD). Each sample contains four sensitive features—Age Group, Region, Gender, and Color—along with six non-sensitive features independently sampled from a uniform distribution within a value range $[0, 1]$. The sensitive features are generated as follows: (1) Gender is uniformly drawn from {*Female, Male*}; (2) Color is uniformly drawn from {*Red, Blue*}; (3) Age Group is drawn from {*Child, Youth, Middle, Elder*} with equal probability; (4) Region follows a fixed cyclical sequence: Asia, Europe, Africa, America, Oceania.

We then generate true labels $Y$ for the dataset, where diagnosis is more challenging for a specific subgroup defined by the Exclusive NOR (XNOR) operation (cf. Appendix B.1). Specifically, we assume $X[0]$ is Color, $X[1]$ is Gender, and $X[2]$ is any non-sensitive feature, and define $Y$ based

---

[1]Our code is publicly available at https://github.com/Xusr1123/FaReG.

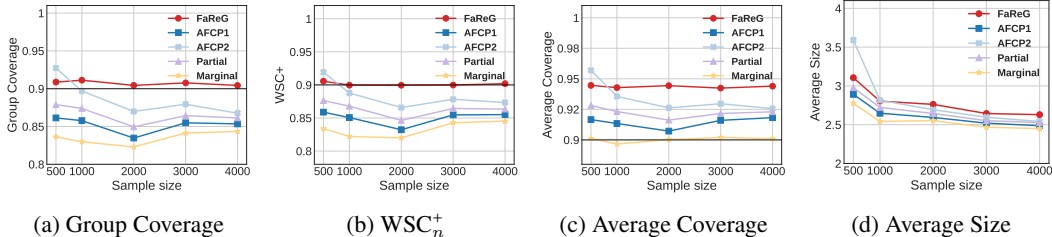

Figure 3: Performance of prediction sets produced by different CP methods on synthetic data w.r.t. the total number of training and calibration data instances. Only our FAREG achieves the ideal conditional coverage (0.9), and meanwhile, does not sacrifice too much information (set sizes) compared to baselines.

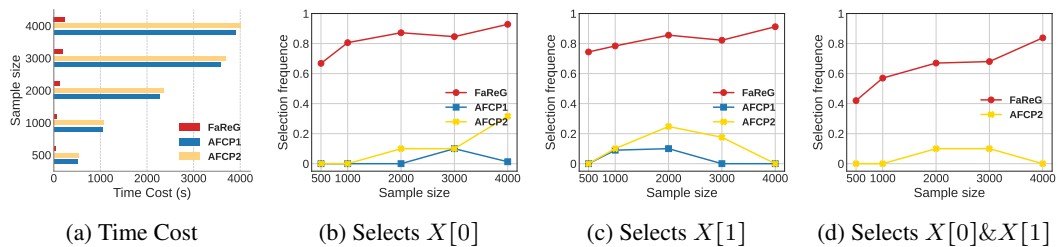

Figure 4: Fig. (a) reports the running time of different CP methods with the increasing total number of training and calibration data instances. Fig. (b)–(d) are the results of the selection frequency of target features $X[0]$ and $X[1]$. As the sample size increases, our method becomes more consistent with target features.

solely on these three attributes. Through the label generation, we have the following subgroup $X[0] \odot X[1]$ = *True*: *Color=Red (True) & Gender=Female (True)* or *Color=Blue (False) & Gender=Male (False)*, simulating a real-world situation that algorithmic biases occur on this subgroup.

Fig. 3 depicts the results of conditional coverage, average coverage (marginal coverage), and average prediction set size (efficiency), respectively. For conditional coverage, Group Coverage is the coverage on the subgroups defined by XNOR operation as mentioned in data construction, and we compute $WSC_n^+$ on four predefined sensitive features. In Fig. 3a and 3b, our FAREG is the only one that always achieves valid coverage (greater than 0.9) for the targeted group with varying sample sizes. Although the conditional coverage of AFCP2 also exceeds 0.9 when the sample size reaches 500, as shown in Fig. 3d, it produces considerably larger prediction sets, which is less informative for decision-making.

In Fig. 4a, we compare the average running time of different CP methods (e.g., the wall-clock time of the entire FAREG pipeline) over 10 repeated experiments, and FAREG significantly reduces the time cost. Actually, the training of the encoder-decoder network occupies most of the time cost (e.g., in one trial with a sample size of 2,000, the training step required 161.3 seconds, while the rest took only 0.8 seconds). Additionally, the training time of the encoder-decoder network scales linearly w.r.t the sample size since we fix the epoch number and batch size in our algorithm. This result is consistent with the analysis in Section 3.2.

To determine which features are selected by our method, we analyze the predictive variable $S$ and the reconstructed feature $\hat{X}$ by perturbing the latent representation $Z$, following the Beta-VAE approach (Higgins et al., 2017). Specifically, we impose a slight perturbation (e.g., ±0.001) on each dimension of $Z$ and identify the dimension that most influences $S$. Given this influential dimension and prior knowledge (as in AFCP2) that there are exactly two target features, we compute the change ratios for each dimension of $\hat{X}$ before and after perturbation, and select the two features with the top-2 maximum change ratios.

Figures 4b, 4c, and 4d respectively report the frequency of selecting $X[0]$ or $X[1]$ individually, and that of selecting both $X[0]$ and $X[1]$ simultaneously. The results demonstrate that our approach

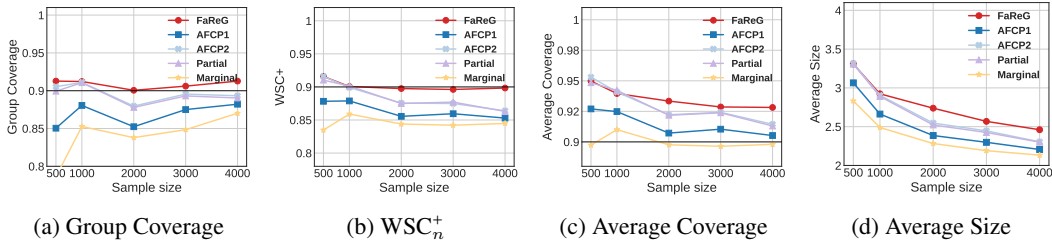

| (a) Group Coverage | (b) WSC$_n^+$ | (c) Average Coverage | (d) Average Size |

Figure 5: Performance of prediction sets produced by different CP methods on the Nursery data w.r.t. the total number of training and calibration data instances. Only our FAREG achieves the ideal conditional coverage (0.9) and keeps most of the uncertainty information of sets.

captures more target features than the baselines, and this advantage becomes more pronounced as the sample size increases.

Additionally, we present the results of parameter sensitivity and group visualization in Appendix B.2 and B.3, respectively.

### 4.3  NURSERY DATA

We evaluate our FAREG and baseline methods on the publicly available Nursery data (Rajkovic, 1989), originally constructed from a hierarchical decision model developed to rank applications for nursery schools. The dataset comprises 12,960 instances, each described by eight categorical features: Parents' occupation (3 levels, *Parent:=\{usual, pretentious, great-pret\}*), Child's nursery (5 levels), Family form (4 levels), Number of children (4 levels), Housing conditions (3 levels), Financial standing (2 levels, *Finance:=\{convenient, inconv\}*), Social conditions (3 levels) and Health status (3 levels). The task is to classify applications into one of five priority ranks. We take all features into account (as sensitive features) except Housing conditions.

In data preprocessing, we strictly follow Zhou & Sesia (2024), and consider a group defined by *Parent=usual* & *Finance=inconv* or *Parent=pretentious* & *Finance=inconv*. To make the issue more interesting and control the degree of algorithmic bias, we corrupt the labels of instances in such a group by adding independent, uniform noise and rounding to the nearest integer (label) as similar as Zhou & Sesia (2024). This perturbation amplifies the intrinsic unpredictability of the group defined before, thereby increasing its vulnerability to algorithmic bias.

Fig. 5 presents the results. Our method consistently achieves the valid coverage under both conditional coverage metrics, i.e., Group Coverage and WSC$_n^+$, outperforming all baselines. Partial and AFCP2 perform better than the other CP methods, but FAREG still achieves superior results.

## 5  RELATED WORK

Conformal Prediction (CP) has seen vigorous development in recent years (Vovk et al., 2005; Smith, 2024). Its applications span diverse domains, from image classification (Sadinle et al., 2019) and object detection (Teng et al., 2023) to large language models (Kumar et al., 2023).

Some CP work, building on the split conformal framework (Papadopoulos et al., 2002; Lei et al., 2018), introduces advanced nonconformity scores to ensure valid marginal coverage on the empirical data distribution. For example, Romano et al. (2019) gives a nonconformity score based on quantile regression, while Romano et al. (2020b) and Angelopoulos et al. (2020) design nonconformity scores for classification. Additionally, Hoff (2023) proposes a nonconformity score to achieve Bayes optimal coverage.

Another line of work has explored various notions of equalized coverage (Romano et al., 2020a) and empirically evaluated the corresponding conformal predictors in real-world applications (Lu et al., 2022). For regression tasks, Wang et al. (2023) guarantees equal coverage rates across more fine-grained groups on continuous features, and Liu et al. (2022) propose to learn a real-valued quantile function with respect to sensitive features. They address a distinct notion of equalized coverage tailored to continuous outcomes. In classification, label-conditional coverage is a common

alternative to equalized coverage (Vovk et al., 2003; Löfström et al., 2015; Ding et al., 2023). This work defines the groups to be protected based on the label $Y_{N+1}$, instead of the features $X_{N+1}$. Jung et al. (2022), Vadlamani et al. (2025), and Gibbs et al. (2025) adopt group-conditional coverage, which is analogous to equalized coverage, to improve prediction sets. Different from the previous work, our approach FAREG can adaptively identify unfairly treated groups without the assumption that such groups are pre-defined. AFCP (Zhou & Sesia, 2024) develops an algorithm to construct CP sets with valid equalized coverage for adaptively selected groups, which establishes the current state-of-the-art for equalized coverage tasks.

## 6 CONCLUSION

In this paper, we propose FAREG, a fair conformal prediction method that learns latent groups to achieve adaptive equalized coverage. By leveraging a variational encoder-decoder to discover subgroups wih poor coverage in a high-level feature space, our approach captures complex algorithmic biases that linear methods may neglect and can be adapted to different fairness notions. We also propose $\text{WSC}^+$, a nonlinear metric for evaluating the conditional coverage of unfair groups more accurately. Extensive experiments confirm that FAREG efficiently offers stronger fairness guarantees, showing a more expressive and practical path toward fair, reliable conformal inference.

**Limitations.** The enhanced expressivity of representation-based groups may sacrifice model interpretability partially, compared to groups explicitly defined on manifest features. However, the encoder-decoder structure compensates this shortcoming well via reconstructing the input $X$, which is empirically confirmed by Section 4.2 and Appendix B.3.

## ACKNOWLEDGMENT

We appreciate anonymous reviewers for their valuable comments. This work is supported by the Frontier Technologies R&D Program of Jiangsu (BF2024059), National Natural Science Foundation of China (Grants #62025202), and the Collaborative Innovation Center of Novel Software Technology and Industrialization. T. Chen is partially supported by overseas grants from the State Key Laboratory of Novel Software Technology, Nanjing University (KFKT2023A04, KFKT2025A05). Yuan Yao is the corresponding author.

## REPRODUCIBILITY STATEMENT

To facilitate the reproducibility of our work, we have made our source code available online through a public repository. All experimental details, including dataset partitions, hyperparameter configurations, and model details, are fully documented in Appendix B.1. We are confident that these materials provide the necessary information to replicate our findings.

## USAGE OF LLMS

Large Language Models (LLMs) were utilized exclusively as writing assistants to enhance the linguistic quality of this manuscript, focusing on improving clarity, grammar, and readability. Their involvement was strictly limited to this editorial function. LLMs played no role in any substantive research components, including conceptualization, experimental design, data analysis, interpretation of results, or scientific content creation. All intellectual contributions, methodological developments, findings, and conclusions originate solely from the authors.

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

# A  TECHNICAL PROOFS

## A.1  VARIATIONAL INFERENCE

As mentioned in Section 3.1, our optimization objective is as follows,

$$\max I(Z, S) + I(Z, X) - \beta I(Z, i).$$

First of all, we consider $I(Z, S)$ and

$$I(Z, S) = \int p_\theta(s, z) \log \frac{p_\theta(s, z)}{p_\theta(s) p_\theta(z)} \, \mathrm{d}s \, \mathrm{d}z = \int p_\theta(s, z) \log \frac{p_\theta(s \,|\, z)}{p_\theta(s)} \, \mathrm{d}s \, \mathrm{d}z. \tag{9}$$

Since the KL divergence between two conditional probability distribution $p_\theta(s \,|\, z)$ and $q_\phi(s \,|\, z)$ is non-negative, we have

$$D_{\mathrm{KL}}(p_\theta(s \,|\, z) \| q_\phi(s \,|\, z)) \geq 0 \Rightarrow \int p_\theta(s, z) \log p_\theta(s \,|\, z) \, \mathrm{d}s \geq \int p_\theta(s, z) \log q_\phi(s \,|\, z) \, \mathrm{d}s,$$

where $q_\phi(s \,|\, z)$ is a variational approximation to the intractable distribution $p_\theta(s \,|\, z)$.

Plugging the above inequality into Eq. 9, we obtain

$$
\begin{aligned}
I(Z, S) &\geq \int p_\theta(s, z) \log \frac{q_\phi(s \,|\, z)}{p_\theta(s)} \, \mathrm{d}s \, \mathrm{d}z \\
&= \int p_\theta(s, z) \log q_\phi(s \,|\, z) \, \mathrm{d}s \, \mathrm{d}z + \int p_\theta(s) \log p_\theta(s) \, \mathrm{d}s \\
&\geq \int p_\theta(s, z) \log q_\phi(s \,|\, z) \, \mathrm{d}s \, \mathrm{d}z,
\end{aligned}
\tag{10}
$$

where the second inequality is derived by the non-negativity of entropy.

Since $S \perp\!\!\!\perp Z \,|\, X$ holds, we have

$$p_\theta(s, z) = \int p_\theta(x, s, z) \, \mathrm{d}x = \int p_\theta(x) p_\theta(s \,|\, x) p_\theta(z \,|\, x) \, \mathrm{d}x.$$

Hence, we get

$$I(Z, S) \geq \int p_\theta(x) p_\theta(s \,|\, x) p_\theta(z \,|\, x) \log q_\phi(s \,|\, z) \, \mathrm{d}x \, \mathrm{d}s \, \mathrm{d}z. \tag{11}$$

Similar to Eq. 10, we also have

$$
\begin{aligned}
I(Z, X) &\geq \int p_\theta(x, z) \log q_\varphi(x \,|\, z) \, \mathrm{d}x \, \mathrm{d}z \\
&= \int p_\theta(x) p_\theta(z \,|\, x) \log q_\varphi(x \,|\, z) \, \mathrm{d}x \, \mathrm{d}z.
\end{aligned}
\tag{12}
$$

As for $I(Z, i)$, we have

$$
\begin{aligned}
I(Z, i) &= \sum_i \int p_\theta(z \,|\, i) p_\theta(i) \log \frac{p_\theta(z \,|\, i)}{p_\theta(z)} \, \mathrm{d}z \\
&= \frac{1}{N} \sum_i \int p_\theta(z \,|\, x_i) \log \frac{p_\theta(z \,|\, x_i)}{p_\theta(z)} \, \mathrm{d}z \\
&\leq \frac{1}{N} \sum_i \int p_\theta(z \,|\, x_i) \log \frac{p_\theta(z \,|\, x_i)}{r(z)} \, \mathrm{d}z,
\end{aligned}
\tag{13}
$$

where $r(z)$ is a variational approximation to the posterior distribution $p_\theta(z)$. We usually set $r(z)$ as a standard normal distribution $\mathcal{N}(0, 1)$ in practice.

Combining Eq. 11 with Eq. 12 and Eq. 13, we obtain

$$
\begin{aligned}
I(Z, S) + I(Z, X) - \beta I(Z, i) &\geq \int p_\theta(x) p_\theta(s \,|\, x) p_\theta(z \,|\, x) \log q_\phi(s \,|\, z) \; \mathrm{d}x \, \mathrm{d}s \, \mathrm{d}z \\
&+ \int p_\theta(x) p_\theta(z \,|\, x) \log q_\varphi(x \,|\, z) \; \mathrm{d}x \, \mathrm{d}z - \frac{\beta}{N} \sum_i \int p_\theta(z \,|\, x_i) \log \frac{p_\theta(z \,|\, x_i)}{r(z)} \, \mathrm{d}z.
\end{aligned}
$$

With Monte Carlo sampling, we use the empirical dataset on $\{X_i, S_i, Y_i\}_{i=1}^N$ to estimate $p_\theta(x)p_\theta(s\,|\,x)$ and $p_\theta(x)$, where $S_i$ is computed by minimizing the conditional coverage of groups defined by $\mathbf{S} = \{s_1, \ldots, s_N\}$, i.e., $\mathbb{P}_n[Y_i \in C(X_i)\,|\,X_i \in \hat{\mathcal{G}}_\mathbf{S}]$ on $\{X_i, S_i, Y_i\}_{i=1}^N$. We leverage the reparameterization trick (Kingma & Welling, 2013) as mentioned in Section 3.1, and finally obtain

$$\mathcal{L} = -\frac{1}{N}\sum_{i=1}^N \left( \mathbb{E}_{\tilde{z}\sim f(x_i, \epsilon)}[\log q_\phi(s_i\,|\,\tilde{z}) + \log q_\varphi(x_i\,|\,\tilde{z})] - \beta D_{\mathrm{KL}}(p_\theta(z\,|\,x_i)\|r(z)) \right).$$

### A.2 PROOF OF PROPOSITION 1

*Proof.* We first present a technical lemma, where $P_n h = \frac{1}{N}\sum_{i=1}^N h(X_i)$ and $Ph = \int h(x)dP(x)$, given an observed dataset $\{X_i, Y_i\}_{i=1}^N$.

**Lemma 1** (Boucheron et al. (2005)). *There exists a numerical constant $C_1$ such that for any $\tau > 0$,*

$$|P_n h - Ph| \le C_1 \left[ \sqrt{\min\{P_n h, Ph\}\frac{VC(h)\log N + \tau}{N}} + \frac{VC(h)\log N + \tau}{N} \right]$$

*holds with probability at least $1 - e^{-\tau}$.*

By this Lemma, we have

$$|P_n(Y \in C(X), X \in \hat{\mathcal{G}}_\mathbf{s}) - P(Y \in C(X), X \in \hat{\mathcal{G}}_\mathbf{s})| \tag{14}$$
$$\le C_1 \left[ \sqrt{\min\{P_n(Y \in C(X), X \in \hat{\mathcal{G}}_\mathbf{s}), P(Y \in C(X), X \in \hat{\mathcal{G}}_\mathbf{s})\}\frac{VC(h)\log N + \tau}{N}} + \frac{VC(h)\log N + \tau}{N} \right].$$

Similarly, we get

$$|P_n(X \in \hat{\mathcal{G}}_\mathbf{s}) - P(X \in \hat{\mathcal{G}}_\mathbf{s})| \tag{15}$$
$$\le C_2 \left[ \sqrt{\min\{P_n(X \in \hat{\mathcal{G}}_\mathbf{s}), P(X \in \hat{\mathcal{G}}_\mathbf{s})\}\frac{VC(h)\log N + \tau}{N}} + \frac{VC(h)\log N + \tau}{N} \right].$$

Then, it remains to show that

$$|P_n(Y \in C(X)\,|\,X \in \hat{\mathcal{G}}_\mathbf{s}) - P(Y \in C(X)\,|\,X \in \hat{\mathcal{G}}_\mathbf{s})|$$
$$= \left| \frac{P_n(Y \in C(X), X \in \hat{\mathcal{G}}_\mathbf{s})}{P_n(X \in \hat{\mathcal{G}}_\mathbf{s})} - \frac{P(Y \in C(X), X \in \hat{\mathcal{G}}_\mathbf{s})}{P(X \in \hat{\mathcal{G}}_\mathbf{s})} \right|.$$

Let $a = P_n(Y \in C(X), X \in \hat{\mathcal{G}}_\mathbf{s}), b = P(Y \in C(X), X \in \hat{\mathcal{G}}_\mathbf{s}), c = (P_n - P)(X \in \hat{\mathcal{G}}_\mathbf{s})$ and $d = P(X \in \hat{\mathcal{G}}_\mathbf{s})$. We can derive $b \le d$, and observe that

$$\left|\frac{a}{c+d} - \frac{b}{d}\right| \le \left|\frac{a}{c+d} - \frac{b-c}{c+d}\right| \le \frac{|a-b|}{c+d} + \frac{|c|}{c+d}. \tag{16}$$

Substitute Eq. 14 and Eq. 15 into Eq. 16, and use $\delta = P_n(X \in \hat{\mathcal{G}}_\mathbf{s})$, we obtain

$$\left| \frac{P_n(Y \in C(X), X \in \hat{\mathcal{G}}_\mathbf{s})}{P_n(X \in \hat{\mathcal{G}}_\mathbf{s})} - \frac{P(Y \in C(X), X \in \hat{\mathcal{G}}_\mathbf{s})}{P(X \in \hat{\mathcal{G}}_\mathbf{s})} \right|$$
$$\le \frac{|P_n(Y \in C(X), X \in \hat{\mathcal{G}}_\mathbf{s}) - P(Y \in C(X), X \in \hat{\mathcal{G}}_\mathbf{s})|}{P_n(X \in \hat{\mathcal{G}}_\mathbf{s})} - \frac{|P_n(X \in \hat{\mathcal{G}}_\mathbf{s}) - P(X \in \hat{\mathcal{G}}_\mathbf{s})|}{P_n(X \in \hat{\mathcal{G}}_\mathbf{s})}$$
$$\le C_3 \left[ \sqrt{\frac{VC(h)\log N + \tau}{\delta N}} + \frac{VC(h)\log N + \tau}{\delta N} \right],$$

which completes the proof. $\qquad\qquad\square$

A.3 Optimization Process of Eq. 4

The projection operation of the PGD algorithm described in Section 3.1 requires solving the following optimization to minimize the $\ell_2$ distance:

$$\min_{v_1,\ldots,v_n} \sum_{i=1}^{N} (v_i - u_i)^2 \quad \text{s.t.} \sum_{i=1}^{N} v_i \geq \delta, \quad v_i \in [0,1] \ i = 1,\ldots,N, \tag{17}$$

where $u_1,\ldots,u_N$ are given and $u_i \in [0,1]$ holds for each $i \in [N]$.

With the above constraints, we compute the Lagrangian as

$$\mathcal{L}(v_i; \lambda_i, \mu_i, \omega) = \sum_{i=1}^{N}(v_i - u_i)^2 + \sum_{i=1}^{N}\lambda_i(-v_i) + \sum_{i=1}^{N}\mu_i(v_i - 1) + \omega(\delta - \sum_{i=1}^{N} v_i),$$

where $\{\lambda_i\}_{i=1}^{N}, \{\mu_i\}_{i=1}^{N}$ and $\omega$ are the Lagrange multipliers. Let the partial derivatives vanish, and we have

$$\frac{\partial \mathcal{L}}{\partial v_i} = 2(v_i - u_i) - \lambda_i + \mu_i - \omega = 0 \Rightarrow 2(v_i - u_i) = \lambda_i - \mu_i + \omega$$

For the complementary relaxation conditions, there are four different cases:

- If $v_i = 0$, constraint $v_i \geq 0$ is activated and we have $\lambda_i \geq 0, \mu_i = 0$;
- If $v_i = 1$, constraint $v_i \leq 1$ is activated and we have $\mu_i \geq 0, \lambda_i = 0$;
- If $0 < v_i < 1$, we have $\mu_i = \lambda_i = 0$ and then $v_i = u_i + \omega/2$;
- If $\sum v_i > \delta$, constraint $\sum v_i \geq \delta$ is not activated and then $\omega = 0$; otherwise, $\omega \geq 0$.

When $\sum u_i \geq \delta$, we have $v_i = u_i$, which is an optimal solution to the minimization problem in Eq. 17.

When $\sum u_i < \delta$, let $v_i = \min(1, u_i + \omega/2)$, where $\omega \geq 0$ and $\sum_{i=1}^{N} \min(1, u_i + \omega/2) \geq \delta$. In this case, we resort $\{v_i\}_{i=1}^{N}$ in descending order, i.e., $v_{(1)} \geq v_{(2)} \geq \cdots \geq v_{(N)}$. Let $k \in [N]$ is the greatest index to satisfy $v_{(k)} + \omega/2 \geq 1$ and $v_{(k+1)} + \omega/2 < 1$. Then, constraint $\sum v_i = \delta$ can be written as

$$k \cdot 1 + \sum_{i=k+1}^{N} (v_{(i)} + \omega/2) = \delta.$$

Hence, we obtain

$$\omega = \frac{2(\delta - k - \sum_{i=k+1}^{N} v_{(i)})}{N - k}.$$

In practice, we can compute $k$ and $\omega$ via traversing the value of $k$ from maximum $N$ to minimum 1.

A.4 Proof of Theorem 1

*Proof.* When making the similar assumption as Theorem 1 in AFCP (Zhou & Sesia, 2024), for each group $\hat{\mathcal{G}}_{\mathbf{s}} \in \{\hat{\mathcal{G}}_{\mathbf{s}_t}\}_{t=1}^{T}$, we can substitute $X_{N+1} \in \hat{\mathcal{G}}_{\mathbf{s}}$ for $\phi(X_{N+1}, \hat{A}(X_{N+1}))$ and $X_{N+1} \in \hat{\mathcal{G}}_{\mathbf{s}}^{o}$ for $\phi(X_{N+1}, \hat{A}^{o}(X_{N+1}))$ as conditions, where $\hat{\mathcal{G}}_{\mathbf{s}}^{o}$ is an imaginary oracle group. Then, according to Theorem 1 (Zhou & Sesia, 2024), we have

$$\mathbb{P}[Y_{N+1} \in C(X_{N+1}) \mid X_{N+1} \in \hat{\mathcal{G}}_{\mathbf{s}}] \geq 1 - \alpha.$$

AFCP assumes that the group selection algorithm can always achieve the oracle group $\hat{\mathcal{G}}_{\mathbf{s}}^{o}$, which means that the algorithm must have enough expressiveness to include $\hat{\mathcal{G}}_{\mathbf{s}}^{o}$ into the candidate group space. However, this necessary condition could be violated, as AFCP's candidate group space is limited to linear groups defined by individual features. In contrast, our method, FAREG, employs a more expressive model that extends its candidate group space into the nonlinear realm. Consequently, the guarantee for FAREG remains valid for groups defined by complex, nonlinear feature combinations.

Next, we formally analyze the expressiveness of AFCP and our FAREG based on the VC-dimension. As described in Section 1, AFCP computes the group coverage scores for each feature and greedily

picks the most sensitive feature with the lowest group coverage score. The essence of such a process is a decision stump dividing all features into two parts (sensitive or not sensitive) using a threshold, and thus its VC-dimension is 2. In contrast, based on established theory (Shalev-Shwartz & Ben-David, 2014), the VC-dimension of FAREG scales with its parameter size $M$, i.e.,

$$\text{VC}(\text{AFCP}) = 2, \quad \text{VC}(\text{FAREG}) = \mathcal{O}(M).$$

Hence, the VC-dimension of our FAREG is typically far larger than that of AFCP, indicating the stronger expressiveness of our method, i.e., our candidate group space serves as a superset of AFCP's candidate group space.

$\square$

### A.5 PROOF OF PROPOSITION 2

*Proof.* According to Proposition 1 and the definition of $\text{WSC}_n^+$ (Eq. 7), we obtain

$$\sup_{\pi \in \Pi}\{|\text{WSC}_n^+(C, \pi) - \mathbb{P}(Y \in C(X) \mid a \le \pi(X) \le b)|\} \le \mathcal{O}(1)\sqrt{\frac{VC(\Pi)\log N}{\delta N}}$$

by omitting $\tau$. Then, we eliminate the absolute value as

$$-\mathcal{O}(1)\sqrt{\frac{VC(\pi)\log N}{\delta N}} \le \text{WSC}_n^+(C, \pi) - \mathbb{P}(Y \in C(X) \mid a \le \pi(X) \le b) \le \mathcal{O}(1)\sqrt{\frac{VC(\pi)\log N}{\delta N}},$$

which holds for all $\pi \in \Pi$. Hence, if $\mathbb{P}(Y \in C(X) \mid a \le \pi(X) \le b) = 1 - \alpha$, we can observe

$$\text{WSC}_n^+(C, \pi) \ge \mathbb{P}(Y \in C(X) \mid a \le \pi(X) \le b) - \mathcal{O}(1)\sqrt{\frac{VC(\pi)\log N}{\delta N}}$$

for any $\pi \in \Pi$.

Next, we only need to prove $\mathcal{O}(d^2) \ge VC(\pi)$. Recall that $VC(\pi)$ denotes the VC-dimension of the binary classifier $\pi$, and $\pi = \mathbf{x}^T\mathbf{W}\mathbf{x} + \mathbf{v}^T\mathbf{x}$ is a quadratic function, where $\mathbf{W} \in \mathbb{R}^{d \times d}$ and $\mathbf{v} \in \mathbb{R}^d$. Therefore, the VC-dimension of $\pi$ is equal to the dimension of its expanded feature space $\mathcal{M} = d(d+1)/2 + d$, i.e., $\mathcal{O}(d^2)$, which completes the proof. $\square$

## B FURTHER EXPERIMENT DETAILS

### B.1 DATASET CONSTRUCTION AND HYPERPARAMETERS

Table 2: Hyperparameters of FAREG.

| DATASET | SYNTHETIC DATA | NURSERY DATA |
|---|---|---|
| MODEL | MLP | MLP |
| NUMBER OF LAYERS | 3 | 3 |
| HIDDEN DIMENSION | [64,32] | [64,32] |
| EPOCH | 2000 | 800 |
| BATCH SIZE | 500 | 500 |
| LEARNING RATE | 0.001 | 0.01 |
| $\beta$ | 2.0 | 0.1 |
| $\delta$ | 0.3 | 0.1 |
| $T$ | 20 | 100 |

For the dataset we use to evaluate two metrics in Section 4.1, only $X[0], X[1]$, and $X[2]$ influence the label $Y$ and we define the conditional distribution $P(Y \mid X)$ as

$$P(Y \mid X) = \begin{cases} \left(\frac{1}{3}, \frac{1}{3}, \frac{1}{3}, 0, 0, 0\right), & \text{if } (X[0] \ge 0.1) \oplus (X[1] \ge 0.1) \text{ and } X[2] < 0.5, \\ \left(0, 0, 0, \frac{1}{3}, \frac{1}{3}, \frac{1}{3}\right), & \text{if } (X[0] \ge 0.1) \oplus (X[1] \ge 0.1) \text{ and } X[2] \ge 0.5, \\ (1, 0, 0, 0, 0, 0), & \text{if not } (X[0] \ge 0.1) \oplus (X[1] \ge 0.1) \text{ and } X[2] < \frac{1}{6}, \\ (0, 1, 0, 0, 0, 0), & \text{if not } (X[0] \ge 0.1) \oplus (X[1] \ge 0.1) \text{ and } \frac{1}{6} \le X[2] < \frac{2}{6}, \\ \vdots \\ (0, 0, 0, 0, 0, 1), & \text{if not } X[0] = (X[0] \ge 0.1) \oplus (X[1] \ge 0.1) \text{ and } \frac{5}{6} \le X[2] \le 1. \end{cases}$$

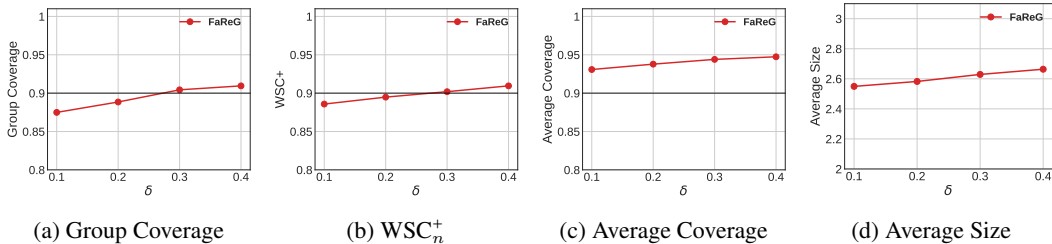

Figure 6: Performance of prediction sets produced by our FAREG on synthetic data w.r.t. the selected group size proportion $\delta$.

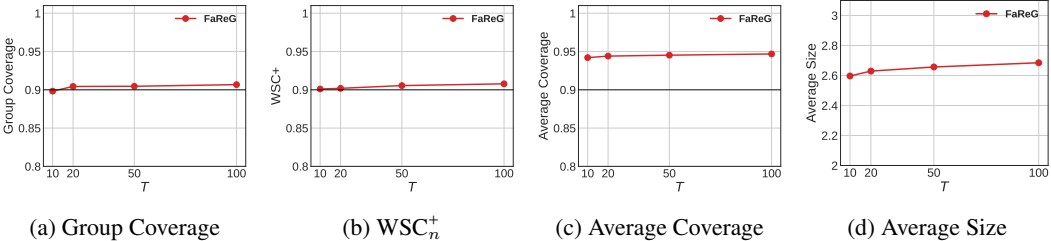

Figure 7: Performance of prediction sets produced by our FAREG on synthetic data w.r.t. the sampling times $T$.

For the classification models as the input of conformal prediction, we strictly follow the settings in (Zhou & Sesia, 2024) on both synthetic and real-world data. To train FAREG to mine unfair groups, we randomly split the calibration set $\mathcal{D}$ into the training set and validation set with the ratio 5:5. We list the hyperparameters of FAREG in Table 2. Note that we use the same network structure for encoders and decoders, i.e., a simple 3-layer MLP, which is consistent with Proposition 1. Since there are three optimization objectives in Eq. 5, which may conflict with each other to some extent, we divide the training into two stages. At the first stage, we train the encoder with the parameter $\theta$ and the decoder with the parameter $\phi$ by fixing the decoder with the parameter $\varphi$ in practice, i.e., the first term $\mathcal{L}_{CC}$ and third term $\mathcal{L}_{KL}$ in Eq. 5. Then, we use $\mathcal{L}_{MSE}$ to reconstruct $X$ based on $Z$ at the second stage.

Recall from Section 4.2 that Color is denoted as $X[0]$, Gender is denoted as $X[1]$, and the first standard feature is denoted as $X[2]$. The conditional distribution of $Y \mid X$ is determined by a simple decision tree, where only $X[0]$, $X[1]$, and $X[2]$ provide valuable predictive information for $Y$, formulated as follows,

$$P(Y \mid X) = \begin{cases} \left(\frac{1}{3}, \frac{1}{3}, \frac{1}{3}, 0, 0, 0\right), & \text{if } X[0] = \textit{Red} \text{ and } X[1] = \textit{Female} \text{ and } X[2] < 0.5, \\ \left(0, 0, 0, \frac{1}{3}, \frac{1}{3}, \frac{1}{3}\right), & \text{if } X[0] = \textit{Red} \text{ and } X[1] = \textit{Female} \text{ and } X[2] \geq 0.5, \\ \left(\frac{1}{3}, \frac{1}{3}, \frac{1}{3}, 0, 0, 0\right), & \text{if } X[0] = \textit{Blue} \text{ and } X[1] = \textit{Male} \text{ and } X[2] < 0.5, \\ \left(0, 0, 0, \frac{1}{3}, \frac{1}{3}, \frac{1}{3}\right), & \text{if } X[0] = \textit{Blue} \text{ and } X[1] = \textit{Male} \text{ and } X[2] \geq 0.5, \\ \left(1, 0, 0, 0, 0, 0\right), & \text{if } X[0] = \textit{Red} \text{ and } X[1] = \textit{Male} \text{ and } X[2] < \frac{1}{6}, \\ \left(0, 1, 0, 0, 0, 0\right), & \text{if } X[0] = \textit{Red} \text{ and } X[1] = \textit{Male} \text{ and } \frac{1}{6} \leq X[2] < \frac{2}{6}, \\ \vdots \\ \left(0, 0, 0, 0, 0, 1\right), & \text{if } X[0] = \textit{Red} \text{ and } X[1] = \textit{Male} \text{ and } \frac{5}{6} \leq X[2] \leq 1, \\ \left(1, 0, 0, 0, 0, 0\right), & \text{if } X[0] = \textit{Blue} \text{ and } X[1] = \textit{Female} \text{ and } X[2] < \frac{1}{6}, \\ \vdots \\ \left(0, 0, 0, 0, 0, 1\right), & \text{if } X[0] = \textit{Blue} \text{ and } X[1] = \textit{Female} \text{ and } \frac{5}{6} \leq X[2] \leq 1. \end{cases}$$

### B.2 PARAMETER SENSITIVITY

In this section, we first investigate the sensitivity of two key parameters: the selected group size proportion $\delta$ and the number of sampling iterations $T$. The results for $\delta$ and $T$ are presented in Fig. 6

Table 3: Performance of prediction sets produced by our FAREG w.r.t the hyperparameter $\beta$.

| $\beta$ | 0.1 | 0.2 | 0.5 | 1.0 | 2.0 |
|---|---|---|---|---|---|
| GROUP COVERAGE | 0.902 | 0.898 | 0.909 | 0.906 | 0.904 |
| AVERAGE SIZE | 2.738 | 2.722 | 2.714 | 2.721 | 2.761 |

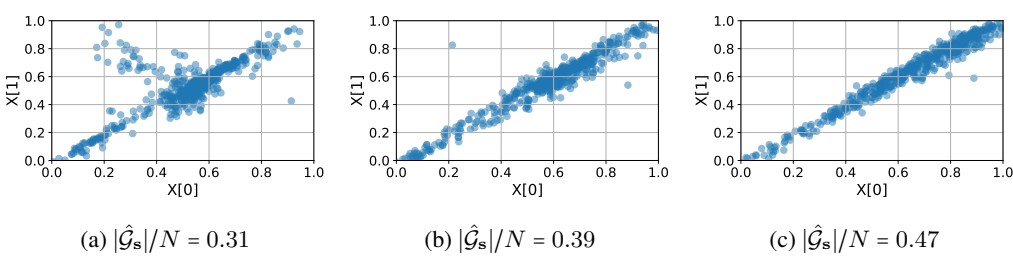

(a) $|\hat{\mathcal{G}}_{\mathbf{s}}|/N = 0.31$        (b) $|\hat{\mathcal{G}}_{\mathbf{s}}|/N = 0.39$        (c) $|\hat{\mathcal{G}}_{\mathbf{s}}|/N = 0.47$

Figure 8: The visualization results of reconstruction $\hat{X}$ when $|\hat{\mathcal{G}}_{\mathbf{s}}|/N$ increases. The latent representation $z$ generally captures an XNOR relationship between $X[0]$ and $X[1]$.

and Fig. 7, respectively. Overall, the metrics Group Coverage and $\mathrm{WSC}_n^+$ show relative insensitivity to the number of sampling iterations $T$, as illustrated in Fig. 7a and 7b. In contrast, both Group Coverage and $\mathrm{WSC}_n^+$ increase with the proportion $\delta$ of the selected group size relative to the entire dataset. This trend empirically provides implicit support for Proposition 1.

We further conduct a sensitivity analysis of the hyperparameter $\beta$ (Eq. 5) on synthetic data with a sample size of 2,000 and report the average results across 10 runs. The experimental results in Table 3 show that, as $\beta$ varies, FAREG consistently achieves valid adaptive equalized coverage (0.9) with relatively steady efficiency.

## B.3 GROUP VISUALIZATION

To analyze how features $X[0]$ and $X[1]$ contribute to group membership in $\hat{\mathcal{G}}_{\mathbf{s}}$, we perturb the encoding $z$ and examine the resulting reconstructions $\hat{X}$. Taking the sample size of 4,000 on the synthetic dataset as an example, we randomly select one run from 10 repeated trials and add perturbations of +0.003 and +0.006 to the fourth dimension of $z$, respectively. The reconstructed features $\hat{X}$ are visualized in Fig. 8.

Fig. 8a shows that, without perturbation, the latent representation $z$ generally captures an XNOR relationship between $X[0]$ and $X[1]$, indicating that the encoder effectively filters out irrelevant feature information. After applying perturbations (see Fig. 8b and 8c), the XNOR pattern becomes more pronounced as $|\hat{\mathcal{G}}_{\mathbf{s}}|/N$ increases, revealing a positive correlation between $X[0] \odot X[1]$ and membership in $\hat{\mathcal{G}}_{\mathbf{s}}$. This result strengthens the interpretability of our approach by demonstrating that the representation-based groups reflect meaningful feature interactions.

## B.4 OTHER SYNTHETIC SETUPS BEYOND XNOR

In this part, we construct a more complicated setup by partitioning the synthetic data in Section 4.2 into eight diverse subgroups: Male Child, Male Youth, Male Middle-Aged, Male Elderly, Female Child, Female Youth, Female Middle-Aged, and Female Elderly. We then impose algorithmic biases, using a procedure analogous to that in Section 4.2, on four of these subgroups: Male Child, Male Youth, Female Middle-Aged, and Female Elderly. In experiments with a sample size of 2,000, we compare our method, FAREG, with the SOTA method AFCP2. The experimental results in Table 4, averaged over 10 runs, also demonstrate that FAREG achieves the ideal conditional coverage (0.9) while sacrificing very little efficiency.

## B.5 CP METHODS WITHOUT ADAPTIVE GROUP SELECTIONS

We next build FAREG upon the results of CondCP (Gibbs et al., 2025), which does not consider the automatic selection of the sensitive groups as discussed in Section 5. As shown in Table 5,

Table 4: Performance of prediction sets produced by different CP methods on the other synthetic setups beyond XNOR.

| METHOD | GROUP COVERAGE | $\text{WSC}_n^+$ | AVERAGE COVERAGE | AVERAGE SIZE |
|---|---|---|---|---|
| AFCP2 | 0.849 | 0.866 | 0.926 | 2.68 |
| FAREG | 0.901 | 0.902 | 0.942 | 2.74 |

experiments on synthetic data with a sample size of 2,000 reveal that CondCP may lack the expressiveness needed to capture complex unfairly treated groups. Consequently, our FAREG method further improves conditional coverage based on CondCP.

Table 5: Performance of prediction sets produced by different CP methods with and without adaptive group selections.

| METHOD | GROUP COVERAGE | $\text{WSC}_n^+$ | AVERAGE COVERAGE | AVERAGE SIZE |
|---|---|---|---|---|
| CONDCP | 0.819 | 0.815 | 0.899 | 2.53 |
| FAREG | 0.904 | 0.899 | 0.944 | 2.76 |
| CONDCP + FAREG | 0.903 | 0.897 | 0.940 | 2.72 |

## B.6 MORE REAL-WORLD DATASETS

In this part, we evaluate our method using the standard ACSIncome dataset from Folktables (Ding et al., 2021), which comprises nine features: Age, Class of worker, Educational attainment, Marital status, Occupation, Place of birth, Relationship, Working hours, Gender, and Race. Among these, Age, Marital status, Sex, and Race are designated as sensitive features. The target variable, i.e., a person's income, is evenly divided into four brackets, forming a 4-class classification task. We further partition the Age feature into four intervals: [17–26, 27–36, 37–46, 47–56]. A specific subgroup is defined by the condition (Age = 17–26 & Sex = Male) or (Age = 47–56 & Sex = Female), into which we manually inject algorithmic biases, following a procedure similar to that in Section 4.3. For time and cost constraints, particularly given the high complexity of the baseline AFCP method, we randomly sample 20,000 instances from the combined datasets of CA, NY, TX, AK, and AL. The proposed FAREG is then compared against the state-of-the-art approach, AFCP2. Experimental results in Table 6 (averaged over 10 independent runs) show that FAREG still consistently achieves the valid conditional coverage of 0.9 while maintaining competitive efficiency.

Table 6: Performance of prediction sets produced by different CP methods on the ACSIncome data.

| METHOD | GROUP COVERAGE | $\text{WSC}_n^+$ | AVERAGE COVERAGE | AVERAGE SIZE |
|---|---|---|---|---|
| AFCP2 | 0.839 | 0.883 | 0.918 | 3.17 |
| FAREG | 0.898 | 0.907 | 0.936 | 3.31 |

