# OpenReview forum: "Fair Conformal Classification via Learning Representation-Based Groups"
_ICLR.cc/2026/Conference — ICLR 2026 Poster_

### Official Review · Reviewer_hizj · 2025-10-27

**Soundness:** 3
**Presentation:** 2
**Contribution:** 3
**Rating:** 6
**Confidence:** 3

**Summary:**

This paper proposes a framework for fair conformal inference in classification tasks. The method adaptively identifies subgroups and enforces conditional coverage guarantees on these learned groups. To evaluate subgroup fairness, the authors introduce a nonlinear variant of the Worst Slab Coverage (WSC) metric, designed to better capture coverage deficiencies in complex or non-linearly defined groups. Experiments are conducted on synthetic data and one real-world dataset.

**Strengths:**

- The approach is **novel**: instead of enumerating or greedily constructing subgroups, the authors learn subgroup structure via representation learning—allowing for more complex group definitions (for instance, XOR-type interactions) that are hard to capture with feature-based heuristics.
- It **improves over the baseline AFCP** in terms of achieving fairer conditional coverage, with only a modest reduction in efficiency (prediction set size). The substantial runtime improvement makes it a much more practical alternative.
- The **methodology is theoretically sound**, supported by clear formulations and proofs that establish its coverage guarantees.

**Weaknesses:**

- The **empirical evaluation is limited**, with most experiments conducted on synthetic data and only a single real-world dataset (Nursery). Including larger and more commonly used datasets—such as the *Folktables* benchmarks used in prior fairness/conditional conformal prediction work ([1]–[3])—would strengthen the experimental evidence.
- The paper does not discuss the **sensitivity of the hyperparameter β** in the final loss function, which may affect the fairness–efficiency trade-off.
- A few parts of the **writing could be polished for clarity**, including the caption for Figure 2/Table 1 and the paragraph starting at line 405.

References

[1] O. Bastani et al: Practical adversarial multivalid conformal prediction [NeurIPS 2022]

[2] C. Jung et al: Batch Multivalid Conformal Prediction [ICLR 2023]

[3] AT. Vadlamani et al: A Generic Framework for Conformal Fairness [ICLR 2025]

**Questions:**

See weaknesses.

- The paper focuses on **equalized coverage** as the fairness notion for conformal prediction. However, other fairness notions—such as those derived from popular ML fairness metrics—have been considered in related work ([3]). Does your method naturally extend to these alternative definitions, and if so, how?

References

[3] AT. Vadlamani et al: A Generic Framework for Conformal Fairness [ICLR 2025]

---

> ### Author Response · Authors · 2025-11-26
>
> We thank Reviewer hizj for the constructive feedback. Your suggestions are very actionable. Here are our responses to your concerns and questions:
>
> > **On Weakness 1 (more real-world datasets)**
>
> As suggested, we have evaluated our method using the standard ACSIncome dataset from Folktables [1], which comprises nine features: Age, Class of worker, Educational attainment, Marital status, Occupation, Place of birth, Relationship, Working hours, Gender, and Race. Among these, Age, Marital status, Sex, and Race are designated as sensitive features. The target variable, i.e., a person’s income, is evenly divided into four brackets, forming a 4-class classification task. We further partition the Age feature into four intervals: [17–26, 27–36, 37–46, 47–56]. A specific subgroup is defined by the condition (Age = 17–26 & Sex = Male) or (Age = 47–56 & Sex = Female), into which we manually inject algorithmic biases, following a procedure similar to that in Section 4.3.  For time and cost constraints, particularly given the high complexity of the baseline AFCP method, we randomly sample 20,000 instances from the combined datasets of CA, NY, TX, AK, and AL. Our proposed method, FAREG, is then compared against the state-of-the-art approach, AFCP2.
>
> Experimental results in the following table (averaged over 10 independent runs) show that FAREG still consistently achieves the valid conditional coverage of 0.9 while maintaining competitive efficiency. These results will be included in the revised version.
>
> |Method   |Group Coverage |WSC+ |Average Coverage |Average Size |
> |:------|:---:|:---:|:---:|:---:|
> |AFCP2|0.839|0.883|0.918|3.17|
> |FAREG|0.898|0.907|0.936|3.31|
>
>
> > **On Weakness 2 (the sensitivity of the hyperparameter β)**
>
> We further conduct a sensitivity analysis of $\beta$ on synthetic data with a sample size of 2000 and report the average results across 10 runs. The experimental results show that, as $\beta$ varies, FAREG consistently achieves valid adaptive equalized coverage (0.9) with relatively steady efficiency. We will include this analysis in the revised version.
>
> |beta   |0.1 |0.2 |0.5 |1.0 |2.0 |
> |:------|:---:|:---:|:---:|:---:|:---:|
> |Group Coverage|0.902|0.898|0.909|0.906|0.904|
> |Average Size|2.738|2.722|2.714|2.721|2.761|
>
> > **On Weakness 3 (the writing issues)**
>
> Thanks for the comment. We have incorporated the suggested revisions and revised our manuscript (as indicated by the red markings in the submitted manuscript).
>
> > **On Question 1 (extendability of FAREG to other fairness notions)**
>
> We believe that FAREG is extendable to fairness notions used in [2]. Specifically, these fairness notions are defined by the coverage gap between under-covered and well-covered groups (smaller is better). To this end, we can extend our approach by simultaneously modeling the distributions for both under-covered and well-covered groups. That is, the model can learn to identify groups with poor coverage alongside those with good coverage. When constructing the final prediction set, we would then actively sample from groups identified as under-covered while avoiding those with good coverage, guided by the learned group distribution. We will discuss it in the revised version.
>
> --------
> ### References
> [1] Retiring Adult: New Datasets for Fair Machine Learning. Ding  et. al, NeurIPS 2021
>
> [2] A Generic Framework for Conformal Fairness. Vadlamani et. al, ICLR 2025

---

### Official Review · Reviewer_9Zja · 2025-10-30

**Soundness:** 3
**Presentation:** 3
**Contribution:** 3
**Rating:** 6
**Confidence:** 3

**Summary:**

This paper proposes FAREG, a conformal prediction method that ensures fairness by adaptively identifying “unfair” subgroups in a learned latent representation space. Unlike prior conformal predictors with equalized coverage that only consider simple or pre-defined groups (e.g. single sensitive features), FAREG learns a representation $Z = f(X)$ via a variational encoder–decoder and discovers complex subgroups (even defined by nonlinear feature combinations like XOR) associated with low coverage. The method then adjusts prediction sets to guarantee adaptive equalized coverage for these discovered subgroups. Additionally, the paper introduces WSC+, a “nonlinear” worst-case conditional coverage metric extending the worst-slab coverage (WSC) metric of Cauchois et al. (2021)

**Strengths:**

The paper tackles the important intersection of uncertainty quantification and algorithmic fairness. Ensuring no subgroup is underserved by predictive uncertainty is crucial for trustworthy AI. The work is timely and relevant to high-stakes domains. FAREG’s combination of representation learning with conformal prediction is novel. It significantly extends prior fair conformal methods (from single-feature groups to rich latent groups).

**Weaknesses:**

- The proposed algorithm is complex, involving a custom VAE training and Monte Carlo sampling of groups. Some steps (the PGD projection and the final aggregation of prediction sets) are not explained in depth in the main text.

- FAREG focuses on one subgroup (or a mixture of one) to protect. If there are multiple distinct biased subgroups, it’s unclear if FAREG can handle them simultaneously.

- some relevant baselines and related works are not mentioned. for instance  (https://arxiv.org/pdf/2505.16115) or (https://arxiv.org/abs/2305.12616)

**Questions:**

- Could you elaborate on how exactly the $T$ samples of $S$ (Algorithm 1, lines 12–16) are used to form the final prediction set $C(X_{N+1})$?


- Under what assumptions does FAREG guarantee $P(Y \in C(X) \mid X \in \hat G) \ge 1-\alpha$ for the discovered group $\hat G$? Is this guarantee exact finite-sample (with sample splitting) or only asymptotic/high-probability?

---

> ### Author Response · Authors · 2025-11-26
>
> We sincerely thank Reviewer 9Zja for their insightful comments. We address your concerns below.
>
> > **On Weakness 1 (the explanation of certain steps)**
>
> Thanks for the comment. We have improved the description of the two mentioned steps (the PGD projection and the final aggregation of prediction sets) in the submitted manuscript (marked in red). Specifically, we have enhanced the description of the PGD projection by adding a comprehensive explanation of its complete algorithmic process. Furthermore, we have refined Section 3.2 to provide a more thorough explanation of the final aggregation procedure. Please refer to the corresponding paragraphs (Lines 224 and 243) for more details.
>
>
> > **On Weakness 2 (simultaneously handling multiple distinct biased subgroups)**
>
> We would like to clarify that FAREG is inherently capable of simultaneously handling  multiple distinct biased subgroups. The reason is that FAREG is designed to adaptively identify unfairly treated subgroups characterized by low coverage directly from raw features, without prior knowledge of any subgroup number or definitions.
>
> Furthermore, we stratify the synthetic data from Section 4.2 into eight different subgroups: Male Child, Male Youth, Male Middle-Aged, Male Elderly, Female Child, Female Youth, Female Middle-Aged, and Female Elderly. Algorithmic biases, following a similar procedure to Section 4.2, are then introduced for four of these subgroups: Male Child, Male Youth, Female Middle-Aged, and Female Elderly. The results in the following table (2000 samples of 10 runs) shows that our FAREG method consistently achieves the target conditional coverage of 0.9 across all subgroups, while sacrificing only slight efficiency. We will include this result in the revised version.
>
> |Method   |Group Coverage |WSC+ |Average Coverage |Average Size |
> |:------|:---:|:---:|:---:|:---:|
> |AFCP2|0.849|0.866|0.926|2.68|
> |FAREG|0.901|0.902|0.942|2.74|
>
>
> > **On Weakness 3 (missing related work)**
>
> Thanks for providing two valuable references [1,2] to us.
>
> For [1], they explore how to improve the conformal fairness, given predefined subgroups. However, traversing all plausible subgroups is often statistically and computationally infeasible in practice, particularly for multi-dimensional (continuous) features. In contrast, our work moves beyond specified groups by adaptively learning representation-based groups from the data. We will discuss this in the revised version.
>
> For CondCP [2], we have discussed this work in the related work (see Line 481). To empirically validate our method, we conduct a comparative experiment on synthetic data with a sample size of 2000. The following results, averaged over 10 runs, demonstrate that FAREG significantly outperforms CondCP on both Group Coverage and WSC+ metrics. This improvement is a direct benefit of the strong expressiveness afforded by our method's ability to learn representation-based groups.
>
> |Method   |Group Coverage |WSC+ |Average Coverage |Average Size |
> |:------|:---:|:---:|:---:|:---:|
> |CondCP|0.819|0.815|0.899|2.53|
> |FAREG|0.904|0.899|0.944|2.76|
>
> > **On Question 1 (the use of $T$ samples to form the final prediction set)**
>
> After obtaining the predictive joint Bernoulli distribution B, which models the probability of each sample belonging to the unfair group $\hat{G}$ (we omit \hat for brevity in the following), FAREG repeats the following process for T iterations: it randomly samples an indicator vector $s_t$ (where $t \in [T]$) from B. Each vector $s_t$ corresponds to a distinct instantiation of the selected group, $G_{s_t}$. For a test point $X_{N+1}$, the method then uses each $G_{s_t}$ as a calibration set to construct a prediction set $C_\mathfrak{m}(X_{N+1}, G_{s_t})$ via the standard conformal prediction paradigm. Finally, these $T$ sets are aggregated into a single, final prediction set $C(X_{N+1})$ using Eq. 7.
>
> > **On Question 2 (the assumptions of FAREG)**
>
> Thank you for raising this question. As with AFCP, FAREG's guarantee relies on the exchangeability assumption, which is common in conformal prediction [3]. Under this assumption, the guarantee holds exactly, independent of the sample size.
>
> -------
> ### References
> [1] A Generic Framework for Conformal Fairness. Vadlamani et. al, ICLR 2025
>
> [2] Conformal prediction with conditional guarantees. Gibbs et. al, Journal of the Royal Statistical Society Series B: Statistical Methodology 2025
>
> [3] Algorithmic learning in a random world. Vovk et. al, Springer 2005

---

### Official Review · Reviewer_LJHP · 2025-10-31

**Soundness:** 3
**Presentation:** 2
**Contribution:** 2
**Rating:** 4
**Confidence:** 4

**Summary:**

The paper introduces FAREG, a fair conformal classification method that finds and fixes under-covered subgroups so that they reach the target coverage, without blowing up prediction-set size. Here are the mains steps of the method.
1. Learn a compact representation Z of the features X.
2. On this Z, train a small classifier that scores how likely each sample belongs to an “unfair subgroup” (i.e., tends to be under-covered).
3. Encourage this classifier to pick the samples with the lowest conditional coverage while keeping the subgroup at least a fraction δ of the data (for statistical reliability).
4. Build a standard conformal prediction set for everyone, then build extra subgroup-specific sets for the selected groups, and take the union. This guarantees adaptive equalized coverage for those groups.

**Strengths:**

1. The problem is meaningful and important.
2. The method appears technically sound.
3. Although the full pipeline is more involved than the summary above, the paper is mostly readable.

**Weaknesses:**

First, the goal of this work is a subset of existing objectives. Prior work has focused on conditional coverage
$$
\mathbb{P}\!\left(Y \in C(X_{n+1}) \mid X_{n+1}=x\right)=1-\alpha,
$$
which is stronger than
$$
\mathbb{P}\!\left(Y \in C(X_{n+1}) \mid X_{n+1}\in \widehat{\mathcal{G}}\right)=1-\alpha.
$$
If a conformal method attains (approximate) conditional validity, the proposed method may be less useful. It would be helpful to show that results still improve when FAREG is built on top of conditionally valid conformal methods such as~[1]. Second, based on Figures 2 and 5, gains in coverage appear to come at the cost of efficiency: the proposed method achieves higher coverage but also larger prediction sets.

[1] Isaac Gibbs, John J Cherian, and Emmanuel J Cand`es. Conformal prediction with conditional guarantees. Journal of the Royal Statistical Society Series B: Statistical Methodology, pp. qkaf008, 2025.

**Questions:**

What is the hypothesis class $\mathcal{H}$ in the proposed algorithm? What is the VC dimension of $\mathcal{H}$ in this setting?

---

> ### Author Response · Authors · 2025-11-26
>
> We thank Reviewer LJHP for their insightful feedback and address your concerns below.
>
> > **On Weakness 1 (usefulness upon conditional CF method)**
>
> We agree that if a conformal method attains (near-) perfect conditional validity, our method would be less useful. However, it is hard to achieve fully conditional validity [1], and existing work is an approximation of perfect conditional coverage to varying degrees (e.g., label-condition in classification [2]). Following your suggestion, we build FAREG upon the results of CondCP [3]. As shown in the following table, experiments on synthetic data with a sample size of 2000 reveal that CondCP may lack the expressiveness needed to capture complex unfairly treated groups. Consequently, our FAREG method further improves conditional coverage based on CondCP. We will include this result in the revised version.
>
>
> |Method   |Group Coverage |WSC+ |Average Coverage |Average Size |
> |:------|:---:|:---:|:---:|:---:|
> |CondCP|0.819|0.815|0.899|2.53|
> |CondCP+FAREG|0.903|0.897|0.940|2.72|
>
>
> > **On Weakness 2 (gains in coverage appear to come at the cost of efficiency)**
>
> Indeed, we would like to clarify that there exists a potential tradeoff between efficiency and coverage [1]. As shown in Figures 3 and 5, our FAREG is the only method that achieves valid adaptive equalized coverage, at the cost of extra efficiency comparable to AFCP2 (see Figure 3).
>
>
> > **On Question 1 (the hypothesis class and its VC dimension)**
>
> Since the goal of FAREG is to determine whether a sample belongs to the unfair group, it can be viewed as a binary classifier denoted as $h$ within a hypothesis class $\mathcal{H}$. As established in existing work [4], the VC-dimension of a neural network with real-valued weights is upper-bounded by $\mathcal{O}(|M|)$, where $|M|$ represents the number of parameters in the network. In practice, we choose a neural network with a moderate number of parameters.
>
> -----------
> ### References
> [1] Conditional validity of inductive conformal predictors. Vovk et. al, JMLR 2012
>
> [2] Class-conditional conformal prediction with many classes. Ding et. al, NeurIPS 2023
>
> [3] Conformal prediction with conditional guarantees. Gibbs et. al, Journal of the Royal Statistical Society Series B: Statistical Methodology 2025
>
> [4] Understanding machine learning: From theory to algorithms. Shai et. al, Cambridge university press 2014

---

### Official Review · Reviewer_eNWr · 2025-11-01

**Soundness:** 2
**Presentation:** 3
**Contribution:** 3
**Rating:** 4
**Confidence:** 3

**Summary:**

The authors introduce FAREG, a method to employ variational information bottleneck to achieve fair conformal prediction by identifying worst performing subgroups. They also propose an improvement over the WSC metric to capture non-linear slabs. They perform experiments on a self-constructed synthetic dataset and the Nursery dataset, comparing against standard baselines. The authors also provide code supporting the reproducibility of their method.

**Strengths:**

- **[S1]** The authors do a great job of motivating the paper well. Figure 1 is a great example of how the current set of conformal prediction methods struggle in certain data settings. There is a clear gap in the literature that needs to be addressed and the manuscript is an attempt at that.
- **[S2]** The core idea of FAREG is quite novel, I haven’t come across works using variational information bottlenecks in this context and it feels like a natural fit for potentially non-linear subgroups.
- **[S3]** The synthetic experiment in Section 4.2 is designed perfectly to highlight the proposed method's strength compared to the baselines and is a great way to show the applicability of the approach.
- **[S4]** The code seems succinct and sufficient, supporting the neat empirical claims in the paper.

**Weaknesses:**

- **[W1]** Figure 4(a) and the claims in Line 374 that the time complexity of FAREG is linear in the number of data instances seems misleading, given that this does not account for the training of the encoder-decoder network which is a non-trivial computation. ACP is a post-processing algorithm which does not involve training of any such network and therefore a fairer comparison would be to compute the wallclock times of the entire algorithms from start to end.
- **[W2]** While the synthetic experiment is a great demonstration of the algorithm's strength, it feels like the perfect application in terms of the XNOR-like bias but the authors do not include any other synthetic constructions that could help understand the algorithm's utility in general situations.
- **[W3]** It’s unclear to me how the following statements all hold true together: Line 87: The interpretability is enhanced; Line 483: Expressivity may sacrifice interpretability; Line 941: This result strengthens the interpretability. Intuitively, it seems that the 2nd statement in the limitations is true, so what do the other two statements mean?
- **[W4]** The proof for Theorem 1 in Section A.4 is extremely handwavy. It uses the Theorem 1 from (Zhou & Sesia, 2024) but never establishes how it is better than it formally. There’s a strange argument about the VC Dimension and higher expressivity written in words, but no formal proof of the statistical consequences on learning the groups on the same data used for calibration.

**Questions:**

- **[Q1] ** See [W1]. Can you provide a more detailed and fair comparison of the wallclock time of FAREG and AFCP, showing the entire training, processing and postprocessing steps.
- **[Q2]** See [W2]. Would be curious to see applications of FAREG on other synthetic setups, which are slightly more complicated than XNOR for example.
- **[Q3]** See [W3]. Could you clarify the confusion about the interpretability of the algorithm and state what the effect of the encoder-decoder architecture is?
- **[Q4]** See [W4]. Could you provide a more formal proof for Theorem 1, clearly highlighting the superiority over AFCP and the effects of that?

**Details Of Ethics Concerns:**

N.A.

---

> ### Author Response · Authors · 2025-11-26
>
> We thank Reviewer eNWr for the constructive feedback. We address your concerns as follows.
>
> > **On Weakness 1 & Question 1 (the wallclock time of FAREG)**
>
> Thanks for pointing this out. We would like to clarify that the reported results in Figure 4(a) already included the wall-clock time over the entire FAREG pipeline. Indeed, the  training of the encoder-decoder network occupies most of the time cost (e.g., in one trial with a sample size of 2,000, the training step required 161.3 seconds, while the rest took only 0.8 seconds). Additionally, the training time of the encoder-decoder network scales linearly w.r.t the sample size since we fix the epoch number and batch size in the algorithm. We will make these descriptions more precise in the revised version.
>
> > **On Weakness 2 & Question 2 (more complicated synthetic setups)**
>
> As suggested, we construct a more complicated setup by partitioning the synthetic data in Section 4.2 into eight diverse subgroups: Male Child, Male Youth, Male Middle-Aged, Male Elderly, Female Child, Female Youth, Female Middle-Aged, and Female Elderly. We then impose algorithmic biases, using a procedure analogous to that in Section 4.2, on four of these subgroups: Male Child, Male Youth, Female Middle-Aged, and Female Elderly. In experiments with a sample size of 2000, we compare our method, FAREG, with the SOTA method AFCP2. The experimental results, averaged over 10 runs, also demonstrate that FAREG achieves the ideal conditional coverage (0.9) while sacrificing very little efficiency. These results will be included in the revised version.
>
> |Method   |Group Coverage |WSC+ |Average Coverage |Average Size |
> |:------|:---:|:---:|:---:|:---:|
> |AFCP2|0.849|0.866|0.926|2.68|
> |FAREG|0.901|0.902|0.942|2.74|
>
> > **On Weakness 3 & Question 3 (the confusion about the interpretability)**
>
> Thanks for pointing this out. We would like to clarify this issue as follows, and will revise the manuscript accordingly.
> Directly learning unfair groups can entail a partial trade-off in model interpretability (Line 483). To mitigate this issue, we have integrated an extra decoder within the Variational Information Bottleneck (VIB) framework, designed to reconstruct the original input feature X from its encoding Z. This helps to *mitigate the interpretability issue* caused by the adopted learning-based method (Lines 87 and 941), supported by [1,2].
>
> Additionally, the process of learning representation-based groups can be viewed as a form of feature selection, which requires the model to filter out non-essential information. The encoder-decoder architecture of the VIB framework is inherently well-suited to this objective, providing a second compelling reason for its application in our method.
>
> > **On Weakness 4 & Question 4 (Theorem 1)**
>
> We would like to clarify the key difference between our theorem and that of AFCP: the guarantee of AFCP only considers *linear* groups, while our FAREG extends its guarantee to the *nonlinear* group space via enhancing expressiveness. Then, we formally analyze the expressiveness of AFCP and FAREG based on the VC-dimension. We have updated the relevant proof (Line 802, marked in red) in the submitted manuscript.
>
>
> --------------------------------------------------------------------------------------
> ### References
> [1] beta-VAE: Learning Basic Visual Concepts with a Constrained Variational Framework. Higgins et. al, ICLR 2017
>
> [2] Isolating Sources of Disentanglement in VAEs. Ricky et. al, NeurIPS 2018

---

### Author Response · Authors · 2025-11-26
**Revision Summary**

​​Dear ICLR 2026 AC:


We sincerely thank all reviewers for their insightful comments and constructive feedback, which have been valuable in helping us improve the paper. We appreciate their recognition of our work, such as the *important meaning of our research problem* (Reviewer eNWr, LJHP, 9Zja), *the novelty of our method* (Reviewer eNWr, 9Zja, hizj), and *the significant effectiveness improvement* (Reviewer eNWr, 9Zja, hizj). In the following, we provide a summary of the key questions raised by the reviewers as well as our responses/revisions made to address them.

**Clarifications (W1 & Q1, W3 & Q3 of Reviewer eNWr)**. We clarify some misunderstandings/statements about the wallclock time and model interpretability.

**More experiments**. We have included several experiments based on the reviewers’ comments.

1. *more complex setup (W2 & Q2 of Reviewer eNWr, W2 of Reviewer 9Zja)*. We show that the proposed method demonstrates consistently effective performance in a more complex, realistic setup.
2. *addtional baseline (W1 of Reviewer LJHP, W3 of Reviewer 9Zja)*. We include ConfCP as a new baseline, which pursues (approximate) conditional validity given pre-defined groups (our method does not make this assumption). The results show that our method still outperforms ConfCP, since ConfCP lacks enough expressiveness to capture complex unfair groups.
3. *more real-world dataset (W1 of Reviewer hizj)*. We incorporate an empirical evaluation on the Folktables dataset.
4. *parameter sensitivity (W2 of Reviewer hizj)*. We perform a sensitivity analysis for the hyperparameter $\beta$.

**Theoretical analysis (W4 & Q4 of Reviewer eNWr)**. We highlight the superiority of our method over the existing work through a formal VC-dimension analysis in Theorem 1.

**Writings (W1 of Reviewer 9Zja, W3 of Reviewer hizj)**. We polish imprecise statements for clarity throughout the paper.

Thank you for your hard work and support.

Best wishes,

Authors of Paper 24391

---

### Meta-Review · Area_Chair_PUvz · 2026-01-07

**Summary:**

This paper developed a fair conformal classification model by combing representation learning and confirm prediction. Both empirical validation and theoretical guarantees are provided to demonstrate the performance of the proposed approach.

All reviewers acknowledge the proposed approach. The main concerns lie in the lack of clarity in some details and insufficient empirical evaluation. The authors provided more explanation and additional experiments to address these concerns.

Overall, the concerns have been largely addressed in the rebuttal and do not weaken the novelty or contributions of the proposed approach. Therefore, I recommend acceptance.

**Reviewer Concerns:**

The main concerns lie in the lack of clarity in some details and insufficient empirical evaluation. The authors provided more explanation and additional experiments to address these concerns in the rebuttal.

**Reviewer Scores:**

There is no follow-up discussion in the rebuttal phase.

---

### Decision · Program_Chairs · 2026-01-26

Accept (Poster)